# Bacterial outer membrane proteins assemble via asymmetric interactions with the BamA β-barrel

Matthew T. Doyle [ID] [1] & Harris D. Bernstein[1]

The integration of β-barrel proteins into the bacterial outer membrane (OM) is catalysed by the β-barrel assembly machinery (BAM). The central BAM subunit (BamA) itself contains a β-barrel domain that is essential for OM protein biogenesis, but its mechanism of action is unknown. To elucidate its function, here we develop a method to trap a native *Escherichia coli* β-barrel protein bound stably to BamA at a late stage of assembly in vivo. Using disulfide-bond crosslinking, we find that the first β-strand of a laterally 'open' form of the BamA β-barrel forms a rigid interface with the C-terminal β-strand of the substrate. In contrast, the lipid-facing surface of the last two BamA β-strands forms weaker, conformationally hetero-geneous interactions with the first β-strand of the substrate that likely represent intermediate assembly states. Based on our results, we propose that BamA promotes the membrane integration of partially folded β-barrels by a 'swing' mechanism.

---

[1] Genetics and Biochemistry Branch, National Institute of Diabetes and Digestive and Kidney Diseases, National Institutes of Health, Bethesda, MD 20892, USA. Correspondence and requests for materials should be addressed to H.D.B. (email: harris_bernstein@nih.gov)

                                                                                                                                1

The outer membrane (OM) of Gram-negative bacteria is a key load-bearing structure[1] that is densely packed with proteins[2]. Integral outer membrane proteins (OMPs) serve multiple critical cellular functions, including nutrient uptake[3], protein secretion[4,5] and adhesion[6]. Almost all of these proteins are anchored to the OM by a unique 'β-barrel' structure. OMP β-barrels form closed antiparallel β-sheets held together by hydrogen bonds between the first β-strand and a conserved C-terminal β-strand (the 'β-signal') that create a 'β-seam'[7–9] (Fig. 1a, top). The result is a highly stable amphipathic structure ranging in size from 8 to 36 β-strands with alternating hydrophobic lipid-facing and hydrophilic lumen-facing residues. Some OMPs contain separately folded extracellular or periplasmic domains or segments that are embedded inside the β-barrel[10]. Proteins that have

a β-barrel architecture are also found in the OM of organelles of bacterial origin including mitochondria and chloroplasts[11,12].

Once OMPs are translocated across the inner membrane by the Sec-translocon and maintained in an assembly-competent state by periplasmic chaperones, they are assembled and integrated into the OM in the absence of external energy inputs[13]. This is achieved by the β-barrel assembly machinery (BAM), which in *E. coli* is composed of a β-barrel protein (BamA) and four accessory lipoproteins (BamBCDE) that associate with the periplasmic POTRA (polypeptide transport-associated) domains of BamA[14]. Both BamA and BamD are conserved and essential for viability[15,16], while *bamB*, *bamC* and *bamE* mutants produce variable OMP assembly defects[10]. BamA is a member of the Omp85 superfamily, a group of 16-stranded β-barrel proteins that

**Fig. 1** MBP−76EspP forms a stable OMP assembly intermediate in vivo. **a** Top, Crystal structure of the EspP β-barrel and embedded linker (PDB id: 3slj)[42]. The β-seam between β-strand 1 (magenta) and the C-terminal β-signal stand (green, β-stand 12) is shown. Bottom, Structure of BamA derived from the cryo-EM structure of the BAM holocomplex (PDB id: 5ljo)[21]. The lateral opening between β-strand 1 (orange) and β-strands 15 (salmon) and 16 (yellow) is shown. **b** Primary structures of MBP−76EspP, 76EspP and HisBamA. SS: signal sequence; TS: TwinStrepII-tag; MBP: maltose-binding protein; Pass.: passenger domain, PK: proteinase K; H: His₈ tag. Green and white triangles show the location of the surface-exposed loop cleaved by PK and the native intra-barrel cleavage site, respectively. The binding sites of antibodies/antisera used throughout this study are indicated. **c** Model depicting the normal assembly of 76EspP (left) and 'arrest-release' assembly of MBP−76EspP (right). During the assembly of 76EspP, the passenger domain is secreted in a C-to N-terminal direction while the β-barrel is bound to BamA. Although the translocation of the MBP−76EspP passenger domain is initiated properly, the MBP moiety stalls translocation and traps the β-barrel in association with BamA. Assembly can be restarted by adding PK to release an N-terminal fragment that contains the MBP moiety. **d** E. coli BL21(DE3) were transformed with either pMTD607/pMTD372 or pMTD826/pMTD372, and 76EspP or MBP−76EspP was expressed with HisBamABCDE under optimised conditions (see Supplementary Fig. 1). Cells were PK-treated or mock-treated on ice. Immunoblots were then conducted using the indicated antibodies/antisera. A cross-reactive protein is denoted (*). **e** Reactivation of MBP−76EspP assembly ('arrest-release'). BL21 (DE3) that expressed MBP−76EspP and HisBamABCDE were treated with PK (or mock-treated) over a 30 min time course at 25 °C. Immunoblots were then conducted using the indicated antibodies/antisera. Cross-reactive proteins are denoted (*). The fraction of the MBP−76EspP that was completely assembled (% mature β-barrel) was calculated using the blot that was probed with αEspP_βN. Representative results from at least two independent experiments are shown in **d**, **e**. Source data are provided as a Source Data file

have been implicated in both membrane protein insertion and protein secretion reactions in bacteria and organelles[17,18].

Despite multiple solved structures[19–21], how BAM interacts with client β-barrel proteins to catalyse assembly remains the most pressing question in OMP biogenesis[13]. Two frontrunning models of OMP assembly, the 'threading' model (also known as the 'budding' model) and the 'assisted' model, centre on evidence that the seam between the first and last β-strands of the BamA β-barrel (β1 and β16) is structurally unstable and can open laterally (Fig. 1a, bottom). Indeed, the introduction of disulfide-bonds into BamA that prevent lateral opening of the β-barrel creates a lethal phenotype[20,22] and inhibits OMP assembly in vitro[21]. In the threading model, the β-strands of unfolded OMPs are threaded through the BamA β-barrel lumen and inserted sequentially into the plane of the OM through the lateral opening (the lateral 'gate') in BamA[22–24]. A hybrid-barrel is created during the assembly process that collapses when folding is complete. This model is supported by recent evidence that the mitochondrial homolog of BamA, Sam50, forms two aligned interfaces with C-terminal fragments of a mitochondrial β-barrel protein in isolated mitochondria[18]. The discovery of interactions between the C-terminal β-strand of the substrate and Sam50 β1, and the N-terminal β-strand of the substrate and Sam 50 β16, led to the proposal that insertion involves the binding of the β-signal to Sam β1, the opening and expansion of the lateral gate to accommodate an increasing number of β-hairpins, and the eventual release of the full-length protein into the lipid bilayer. The assisted model posits that the combined effects of the wedge-shape of BamA and dynamics of BamA β16 cause membrane disruption and lower the energy barrier for integration of pre-folded OMPs[24–26]. This model is supported by evidence that OMPs begin to fold in the periplasm prior to integration[27–32], that BamA lowers the kinetic barrier for OMP insertion imposed by lipid head groups[33], and that BamA exerts a greater stimulatory effect on the insertion of OMPs into thicker bilayers[34]. In this model, BamA is more passive, and no specific interactions form between the lateral-opening and the client OMP.

Here, we sought to test these models by probing interactions between BamA and an OMP assembly intermediate in vivo. To this end, we design a method to arrest the assembly of a native β-barrel while it remains stably bound to BAM. This tool allows us to precisely map interactions between BamA and the client OMP via intermolecular disulfide-bond formation. We find that the BamA β-barrel forms two dissimilar interfaces with the assembly intermediate that effectively creates a structure that we refer to as an 'asymmetric hybrid-barrel'. This unique structure, however, does not seem to result from the stepwise threading of β-strands

of the client protein from the lumen of the BamA β-barrel into the OM via a lateral gate. Our results strongly suggest an alternative model for BamA function in which it facilitates the transfer of partially folded β-barrels from the periplasm into the OM. This model accounts for the results of previous studies but differs considerably from both the 'threading' and 'assisted' models.

## Results

**Design of a stable OMP assembly intermediate bound to BamA.** To further test the mechanism of BAM-mediated membrane integration of β-barrel proteins, we sought to obtain a detailed map of interactions between BamA and a stable OMP assembly intermediate in vivo. Because most OMPs are assembled rapidly in living cells, it is challenging to isolate a native β-barrel that is stably bound to BAM at a specific stage of assembly. To overcome this problem, we exploited the unique features of 'autotransporters', a class of OMPs that contain an N-terminal extracellular ('passenger') domain in addition to an average size 12-stranded C-terminal β-barrel domain[35]. Autotransporter β-barrel domains are not completely assembled and released from BAM until the passenger domain, which is translocated across the OM in a C- to N-terminal direction via the formation of an intra-barrel hairpin, is fully secreted[27,36,37]. Nevertheless, almost the entire passenger domain can be deleted or replaced by hetero-logous polypeptides without affecting β-barrel assembly[34,38,39]. Based on evidence that the translocation of fully folded hetero-logous polypeptides by autotransporters is often limited by their size[38,39], we hypothesised that the fusion of a large protein to the passenger domain would create a molecular 'knot-in-a-rope' that arrests assembly after the association of the β-barrel domain with BAM and the initiation of translocation. To test this idea we fused the 40 kDa maltose-binding protein (MBP), which normally folds rapidly in the periplasm, to several N-terminally truncated versions of the E. coli O157:H7 autotransporter EspP (Fig. 1b). We chose EspP because the two domains of the protein are separated in an autocatalytic intra-barrel cleavage reaction following the completion of β-barrel assembly[40] (Fig. 1c, left). This cleavage reaction provides an important internal control that enabled us to easily monitor the assembly status of the fusion proteins. Additionally, many aspects of the assembly of EspP have been characterised[27,28,30,34,36,41] and the crystal structure of its β-barrel domain has been solved[40,42].

Consistent with our hypothesis, we found that the MBP-EspP fusions caused assembly arrest after the fusion proteins were targeted to the OM. E. coli were transformed with a low-copy plasmid encoding TwinStrepII-tagged MBP-EspP fusions

controlled by the *rha* promoter[43] and a second plasmid encoding BAM with a His-tagged copy of BamA controlled by an IPTG-inducible promoter[41] (Fig. 1b). In trial experiments we tested the assembly of fusion proteins that contained passenger domain fragments ('linkers') of varied lengths (115, 97, 76, or 59aa) between MBP and the β-barrel domain. The absence of the cleaved β-barrel domain on immunoblots showed that none of the fusion proteins were able to assemble completely (Supplementary Fig. 1a, left). If the fusion proteins were targeted to BAM and passenger domain translocation was initiated, we predicted that the linkers would be exposed on the cell surface in a hairpin conformation and subject to cleavage by added proteinase K (PK) (Fig. 1c, right). As expected, PK digestion released a ~33 kDa C-terminal fragment corresponding to the β-barrel and a small piece of the linker from all but the smallest fusion protein (Supplementary Fig. 1a, left). PK also released multiple ~50 kD N-terminal fragments from the two largest fusion proteins that were detected on immunoblots with an anti-StrepII antibody, but only a single N-terminal fragment from the fusion containing the 76aa-linker. Furthermore, only the fusion that contains the 76aa-linker was resistant to cleavage by endogenous periplasmic proteases. This observation plus the presence of a single N-terminal PK fragment indicates that the fusion is stably trapped at a specific stage of assembly in which the folded MBP moiety remains close to the OM after it reaches BAM (Supplementary Fig. 1a, right). We therefore performed all further experiments with this fusion, which we refer to as $^{MBP-76}$EspP. Note that the final version of $^{MBP-76}$EspP also contains a surface-exposed TEV cleavage-site in the linker (between EspP residues 984/985, Supplementary Fig. 1b) that, while not exploited in this work, is likely to be useful in future studies.

Under optimised $^{MBP-76}$EspP and BAM expression conditions that we used throughout this study (Supplementary Fig. 1c), no mature β-barrel was detected and most of the fusion protein was cleaved into a C-terminal β-barrel-containing fragment and an N-terminal MBP-containing fragment (Fig. 1d). This observation shows that the assembly of the majority of $^{MBP-76}$EspP was stably arrested at steady-state. Under the same conditions, a control protein that lacked the MBP moiety, $^{76}$EspP, was almost all detected as mature β-barrel due to normal rapid assembly (Fig. 1b, d).

If the assembly of $^{MBP-76}$EspP can be restarted after passenger domain translocation stalls, we expected that PK treatment would cleave the surface-exposed loop of the fusion protein, and that the subsequent release of the MBP moiety into the periplasm would facilitate the completion of assembly (Fig. 1c, right). Strikingly, after cells were incubated with PK at 25 °C, ~90% of the fusion protein was fully assembled within 10 min as indicated by the accumulation of the mature β-barrel domain (Fig. 1e). In contrast, $^{MBP-76}$EspP in mock-treated cells remained stable (Fig. 1e). Therefore, arrested $^{MBP-76}$EspP does not fall into a non-productive pathway, but remains in an assembly-competent state that conserves interactions with BAM that occur during normal assembly.

**A stable β-seam forms between BamA(β1) and $^{MBP-76}$EspP (β12).** Having established a model system in which the assembly of a β-barrel protein arrests stably and reversibly, we were able to begin to precisely map interactions between the arrested $^{MBP-76}$EspP assembly intermediate and BAM subunits. Although no direct evidence exists, it has been proposed that the conserved final strand of OMPs, the β-signal, interacts with BamA during assembly[13,44,45]. Based on a recent analysis of interactions between Sam50 and truncated forms of a mitochondrial outer membrane β-barrel in isolated mitochondria[18], we hypothesised

that BamA β-strand 1 [BamA(β1)] interacts with the β-signal of $^{MBP-76}$EspP [$^{MBP-76}$EspP(β12)]. To test this, we used solved structures of BamA[19–21] and the EspP β-barrel[40,42] as a guide to replace pairs of aligned $^{His}$BamA(β1) and $^{MBP-76}$EspP(β12) residues with cysteine (Fig. 2a). We then expressed the modified $^{MBP-76}$EspP(β12) and BAM, and assessed intermolecular proximity by disulfide-bond formation in vivo after the addition of the thiol-specific oxidiser 4-DPS. In these experiments we retained the two native cysteine residues in BamA loop 6 (C690 and C700) that normally form an intramolecular disulfide-bond[46]. We found that these residues are important for loop 6 to fold into a native conformation but do not interfere with intermolecular disulfide-bond formation (Supplementary Fig. 2).

Consistent with our hypothesis, we observed remarkable levels of disulfide-bond formation between aligned lumen-facing cysteines in $^{MBP-76}$EspP(β12) and $^{His}$BamA(β1). After the addition of 4-DPS, 80–90% of $^{MBP-76}$EspP$_{S1299C}$, $^{MBP-76}$EspP$_{R1297C}$, $^{MBP-76}$EspP$_{N1295C}$ and $^{MBP-76}$EspP$_{N1293C}$ formed high molecular-weight adducts with $^{His}$BamA$_{S425C}$, $^{His}$BamA$_{N427C}$, $^{His}$BamA$_{G429C}$, $^{His}$BamA$_{G431C}$, respectively, that were detected by quantitative duplex immunoblots probed with both anti-StrepII and anti-BamA C-terminal antisera (Fig. 2b, c, Supplementary Fig. 3a). These results, as well as the detection of appreciable levels (~5–10%) of spontaneous oxidation in the absence of 4-DPS, suggests the presence of a stable antiparallel inter-barrel interface spanning the OM when the assembly of $^{MBP-76}$EspP arrests. Importantly, the observation that the assembly of oxidised $^{MBP-76}$EspP adducts was completed upon release of the MBP-containing fragment by PK digestion and subsequent disulfide-bond reduction at 25 °C (Supplementary Fig. 4) confirms that the adducts remained assembly competent.

We next obtained evidence that $^{MBP-76}$EspP(β12) and $^{His}$BamA(β1) form a rigid, non-sliding interface during assembly. We found that distally located cysteine-pairs $^{MBP-76}$EspP$_{N1293C}$/$^{His}$BamA$_{S425C}$ and $^{MBP-76}$EspP$_{S1299C}$/$^{His}$BamA$_{G431C}$ failed to form adducts and that disulfide-bond formation was severely diminished when the residues were misaligned by only one register (Fig. 2d, Supplementary Fig. 3b). We also did not observe significant interactions between $^{MBP-76}$EspP(β12) and $^{His}$BamA (β2), $^{MBP-76}$EspP(β12) and $^{His}$BamA(β15/β16), or $^{MBP-76}$EspP (β1) and $^{His}$BamA(β1) (Supplementary Figs. 3d and 5). Because luminal residues in the mature EspP β-barrel cannot form intermolecular disulfide-bonds[40], these results strongly suggest that both β-barrels are in an 'open' state when $^{MBP-76}$EspP(β12) and $^{His}$BamA(β1) interact. Furthermore, when we co-expressed opposite-oriented (luminal vs. lipid-facing) cysteine-pairs to probe the flexibility and secondary structure of the interface, chemically oxidised disulfide-bond levels were typically low (~5%) and no significant spontaneous disulfide-bond was observed (Fig. 2c, e, Supplementary Fig. 3c). The results indicate that there is considerable rigidity between $^{MBP-76}$EspP(β12) and $^{His}$BamA(β1). A higher level of disulfide-bond formation between $^{MBP-76}$EspP$_{Y1298C}$ and $^{His}$BamA$_{S425C}$ (28%) was observed, but is likely explained by the localisation of BamA$_{S425}$ near a known flexible-hinge between the BamA β-barrel and periplasmic POTRA domains[47].

**A conformationally diverse second inter-barrel interface.** A recent examination of interactions between Sam50 and fragments of mitochondrial outer membrane β-barrels showed that a potentially strong second interface forms between the final β-strand of Sam50 [Sam50(β16)] and the N-terminal β-strand of the client, and suggested that an expanding β-sheet enters the OM via the lumen of Sam50[18]. To determine if a similar second

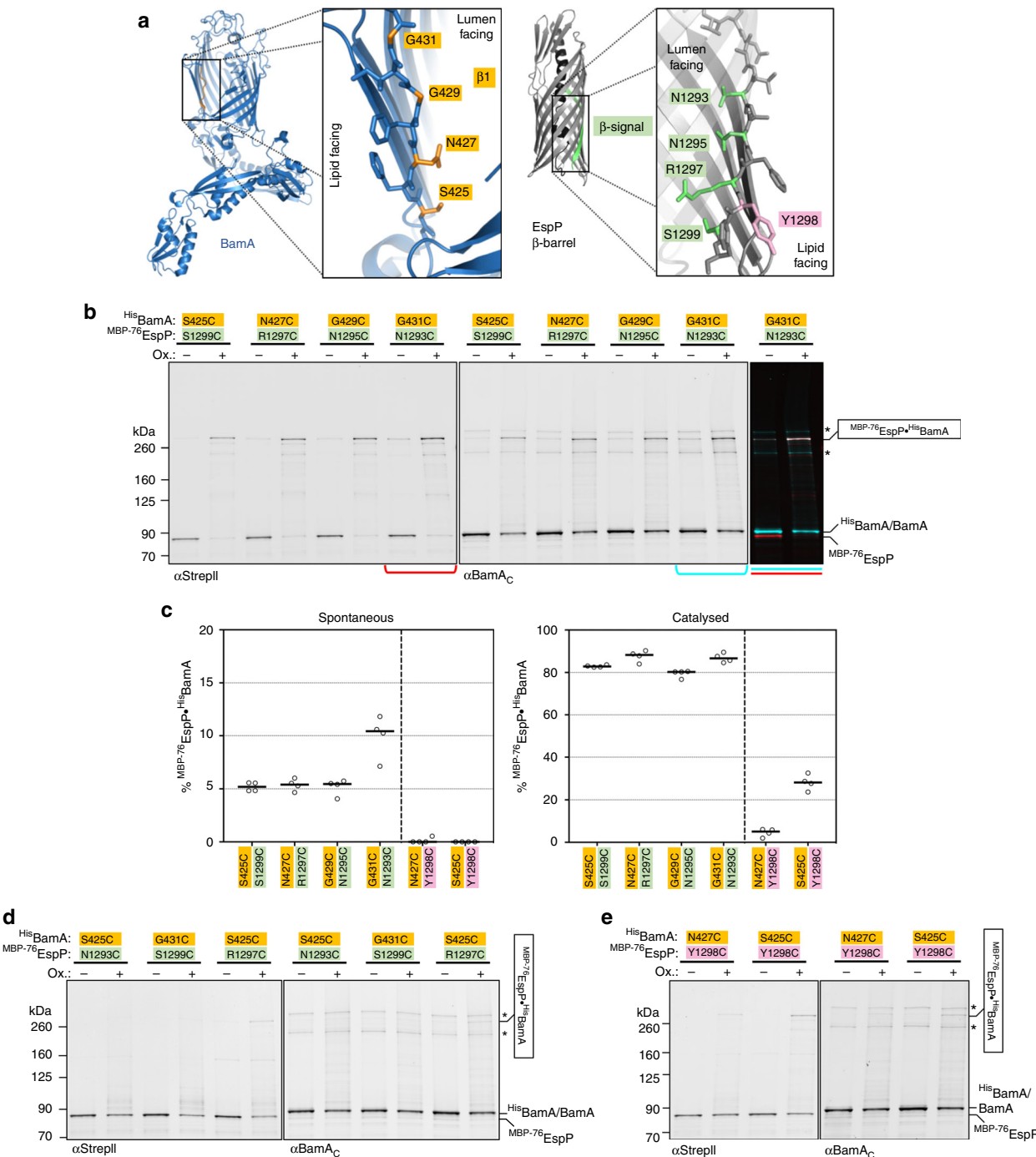

**Fig. 2** During assembly the $^{MBP-76}$EspP(β1) β-signal forms a stable antiparallel seam with BamA(β1). **a** Luminal residues in BamA(β1) (orange), and luminal and lipid-facing residues in $^{MBP-76}$EspP(β12) (green and pink, respectively) are shown. Foreground strands of the $^{MBP-76}$EspP β-barrel are transparent in the zoom box. The 'luminal' and 'lipid-facing' designations are based on the orientation of the residues in solved structures (5ljo, 5dOo, and 3slj)[20,21,42]. The BamA and EspP residue numbers are based on their position in the native protein sequence. **b** BL21(DE3) that expressed $^{MBP-76}$EspP with single cysteine substitutions at a luminal positions in β12 and $^{His}$BamABCDE with cysteine substitutions at a lumen-facing positions in $^{His}$BamA(β1) were mock-treated (−) or treated with 4-DPS (+). Duplex-immunoblots were then conducted using antibodies/antisera against the N-terminus of $^{MBP-76}$EspP (αStrepII, red) and the C-terminus of $^{His}$BamA/BamA (αBamA$_C$, cyan) to monitor disulfide-bond formation between cysteine pairs in vivo. The signals are overlaid for one pair ($^{MBP-76}$EspP$_{N1293C}$/$^{His}$BamA$_{G431C}$, right). Non-specific bands are denoted (*). **c** Quantitation of disulfide-bond formation between $^{MBP-76}$EspP (β12)-$^{His}$BamA(β1) cysteine pairs. Experiments were performed as in **b**, **e** (see below) except that only αStrepII was used for probing immunoblots in mock-treated ('spontaneous') and 4-DPS-treated cells ('catalysed'). Bars = median, N = 4. ANOVA and multiple comparison tests are shown in Supplementary Table 1. **d** The experiments shown in **b** were repeated, except that misaligned lumen-facing cysteine pairs in $^{His}$BamA(β1) and $^{MBP-76}$EspP(β12) were analysed. **e** The experiments shown in **b** were repeated, except that lumen-facing cysteine positions in $^{His}$BamA(β1) and lipid-facing cysteine positions in $^{MBP-76}$EspP(β12) were analysed. Data are representative of at least two independent experiments for **b**, **d** and **e**. Source data are provided as a Source Data file

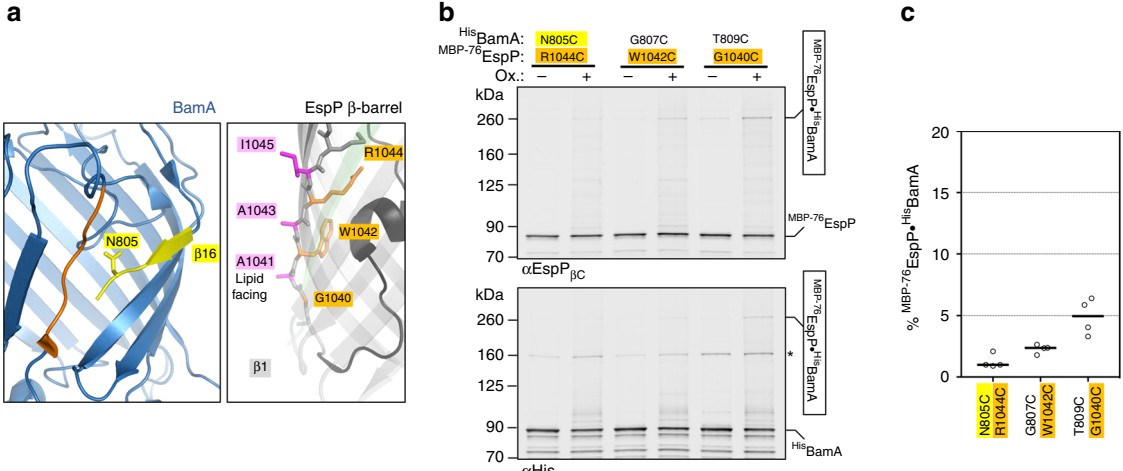

**Fig. 3** $^{MBP-76}$EspP(β1) cysteines bond weakly with luminal-facing cysteines in BamA(β16). **a** Left, view of BamA showing β-strand 16 (yellow) and lumen-facing residue N805. Note that the last several residues of BamA(β16) are not resolved in available structures. Right, view of $^{MBP-76}$EspP(β1) showing lipid-facing (magenta) and lumen-facing (orange) residues. Foreground strands of the EspP β-barrel (including β12, green) are transparent. **b** BL21(DE3) that expressed $^{MBP-76}$EspP with a cysteine substitution at a luminal position in β1 and $^{His}$BamABCDE with a cysteine substitution at a luminal position in $^{His}$BamA(β16) were mock-treated (−) or treated with 4-DPS (+). Duplex-immunoblots were then conducted using antibodies/antisera against the C-terminus of $^{MBP-76}$EspP (αEspP$_{βC}$) and the N-terminus of $^{His}$BamA (αHis) to monitor disulfide-bond formation between cysteine pairs in vivo. Non-specific bands are denoted (*). Data are representative of at least two independent experiments. **c** Quantitation of disulfide-bond formation between $^{MBP-76}$EspP (β1)-$^{His}$BamA(β16) cysteine-pairs in cells that were 4-DPS-treated. Experiments were performed as in **b** except that only αStrepII was used for probing immunoblots. Bars = median, N = 4. ANOVA and multiple comparison tests are shown in Supplementary Table 2. Source data are provided as a Source Data file

interface is formed between inward-facing residues of BamA and assembly-arrested $^{MBP-76}$EspP, we introduced cysteines into luminal positions of $^{His}$BamA(β16) and both lumen- and lipid-facing positions in $^{MBP-76}$EspP(β1) (Fig. 3a). As in our analysis of the $^{MBP-76}$EspP(β12)-$^{His}$BamA(β1) interface, we co-expressed cysteine pairs that we posited would be similarly aligned and oriented. However, the luminal residues $^{MBP-76}$EspP$_{R1044C}$, $^{MBP-76}$EspP$_{W1042C}$, and $^{MBP-76}$EspP$_{G1040C}$ displayed only low levels of chemically-induced disulfide-bond formation (1–5%) with $^{His}$BamA$_{N805C}$, $^{His}$BamA$_{G807C}$ and $^{His}$BamA$_{T809C}$, respectively, based on quantitative immunoblot analysis (Fig. 3b, c, Supplementary Fig. 6a, b). Even lower levels of interaction were detected when the lipid-facing cysteines $^{MBP-76}$EspP$_{I1045C}$, $^{MBP-76}$EspP$_{A1043C}$ and $^{MBP-76}$EspP$_{A1041C}$ were paired with $^{His}$BamA$_{N805C}$, $^{His}$BamA$_{G807C}$ and $^{His}$BamA$_{T809C}$, respectively, or when any of the six $^{MBP-76}$EspP(β1) cysteines were paired in any combination with luminal cysteines in $^{His}$BamA(β16) (Supplementary Fig. 6c–e). Despite the low levels of disulfide-bond formation, $^{His}$BamA$_{T809C}$ consistently produced higher adduct levels regardless of the cysteine location in $^{MBP-76}$EspP(β1). This observation implies significant structural promiscuity and is in line with previous reports of dynamicity and kink formation in BamA(β16)[20,21,25,47]. Nevertheless, the results strongly suggest that BamA does not form two matching interfaces with client proteins.

A close inspection of the BAM cryo-EM structure[21] shows that when BamA is in a 'lateral-open' state, BamA(β15) and BamA(β16) are twisted downward towards the periplasm and curled inward towards the β-barrel lumen. We hypothesised that this unusual conformation might present a unique 'outward-facing' surface that is proximal to the first β-strand of client β-barrels. To test this idea, we substituted cysteine for F785 and I806, two outward-facing amino acids in $^{His}$BamA(β15) and $^{His}$BamA(β16), respectively, that are predicted to reside at a similar membrane depth, and paired them with $^{MBP-76}$EspP(β1) cysteine

substitutions that span the OM (Fig. 4a). A cysteine was also substituted for $^{His}$BamA$_{V784}$ to provide a lumen-facing control.

Consistent with our hypothesis, we observed a significant level of 4-DPS-mediated disulfide-bond formation (typically ~10–20%) between $^{His}$BamA$_{F785C}$ and all of the $^{MBP-76}$EspP(β1) cysteine residues we tested (Fig. 4b, c; Supplementary Fig. 7a, c). There was no clear dependence on the orientation of the $^{MBP-76}$EspP (β1) cysteine residue. Interestingly, cysteines in the middle of the OM spanning segment ($^{MBP-76}$EspP$_{A1043C}$ and $^{MBP-76}$EspP$_{R1044C}$) formed adducts considerably less efficiently than adjacent cysteines. This observation suggests that $^{MBP-76}$EspP(β1) occupies at least two conformational states relative to $^{His}$BamA(β15). In contrast, none of the $^{MBP-76}$EspP(β1) positions formed adducts with $^{His}$BamA$_{V784C}$ (Supplementary Fig. 7d). Thus the outward-facing surface rather than the luminal side of $^{His}$BamA(β15) is in proximity to $^{MBP-76}$EspP(β1). Significant, but less efficient chemically oxidised disulfide-bond formation was also observed between $^{MBP-76}$EspP(β1) cysteines and $^{His}$BamA$_{I806C}$ (typically ~5–15%) that was likewise orientation-independent (Fig. 4d, e, Supplementary Fig. 7b, c). Because we observed consistently lower interactions between $^{MBP-76}$EspP(β1)/$^{His}$BamA(β16) pairs than between $^{MBP-76}$EspP(β12)/$^{His}$BamA(β1) pairs, we checked if any of the $^{MBP-76}$EspP(β1) cysteine mutants affected protein biogenesis. We found that the linkers of all but $^{MBP-76}$EspP$_{G1040C}$ and $^{MBP-76}$EspP$_{W1042C}$ (which contain substitutions at conserved but uncharacterised autotransporter residues) were protease sensitive and therefore properly exposed on the cell surface (Supplementary Fig. 7e). Given that prolonged periplasmic exposure of the EspP β-barrel leads to its degradation, however, the stability of the two mutant proteins, in addition to the consistency of the overall disulfide-bonding pattern, indicate that the G1040C and W1042C substitutions do not affect the engagement of $^{MBP-76}$EspP by BAM. Taken together, these data provide clear evidence for the formation of discrete,

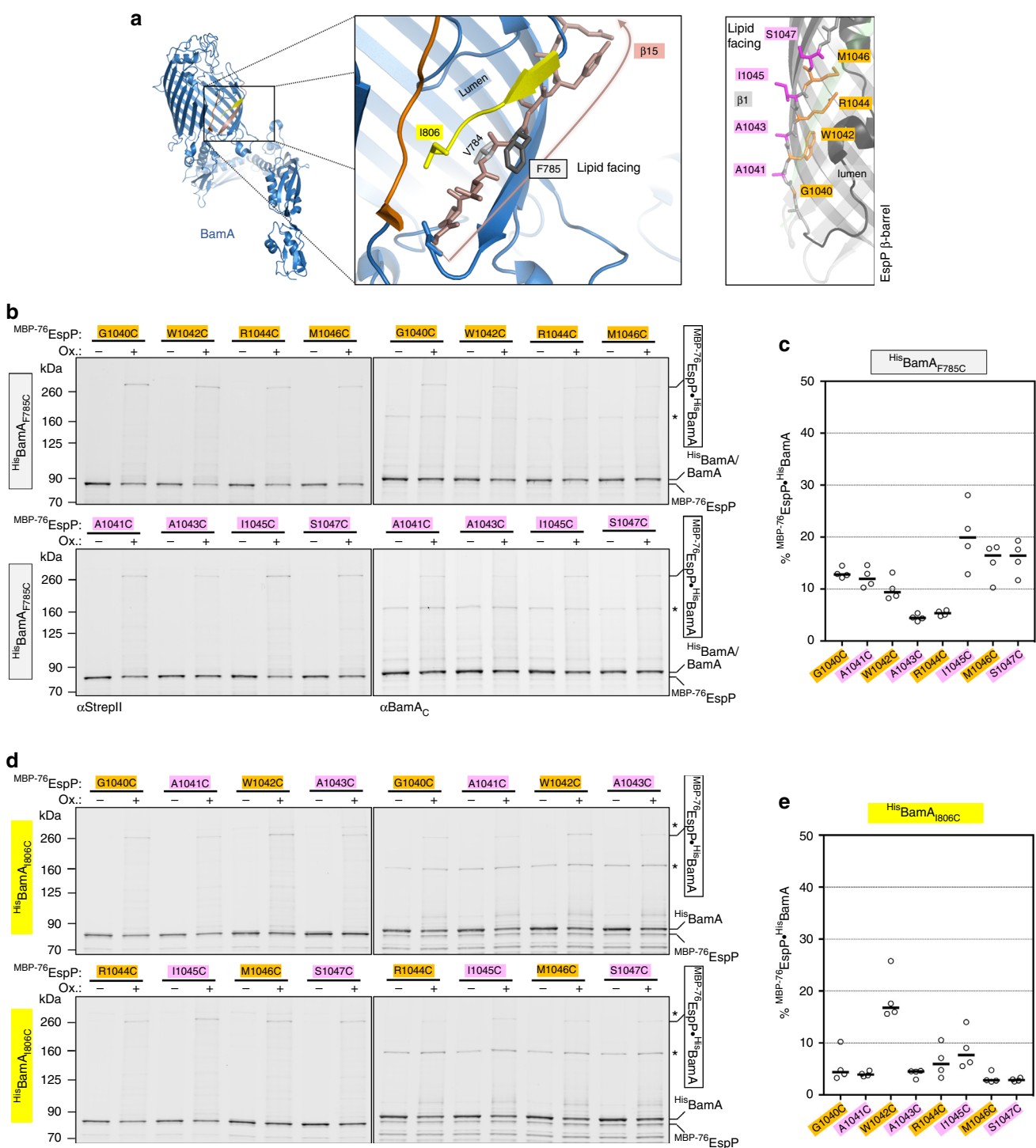

**Fig. 4** Bonding of $^{MBP-76}$EspP(β1) cysteines with lipid-facing cysteines in BamA(β15/16). **a** Left, view of outward-facing residues I806 in BamA(β16) (yellow) and F785 in BamA(β15) (salmon). Right, view of $^{MBP-76}$EspP(β1) showing lipid-facing (magenta) and luminal (orange) residues. Residue S1047 is situated at the interface between the outer leaflet of the OM and the extracellular milieu[54] and is the last membrane embedded residue of β1. **b, d** BL21 (DE3) that expressed $^{MBP-76}$EspP with a single cysteine substitution in β1 and $^{His}$BamABCDE with a $^{His}$Bam$_{I806C}$ (**b**) or $^{His}$BamA$_{F785C}$ (**d**) substitution were mock-treated (−) or treated with 4-DPS (+). Duplex-immunoblots were then conducted using antibodies/antisera against the N-terminus and C-terminus of $^{MBP-76}$EspP (αStrepII or αEspP$_{βC}$) and $^{His}$BamA (αHis or αBamA$_C$) to monitor disulfide-bond formation between cysteine pairs in vivo. Non-specific bands are denoted (*). Data are representative of at least two independent experiments. **c, e** Quantitation of disulfide-bond formation between $^{MBP-76}$EspP (β1) and $^{His}$Bam$_{I806C}$ (**c**) or $^{His}$BamA$_{F785C}$ (**e**) cysteine pairs in 4-DPS-treated cells. Experiments were performed as in **b, d** except that only αStrepII was used for probing immunoblots. Bars = median, N = 4. ANOVA and multiple comparison tests are shown in Supplementary Tables 3 and 4. Source data are provided as a Source Data file

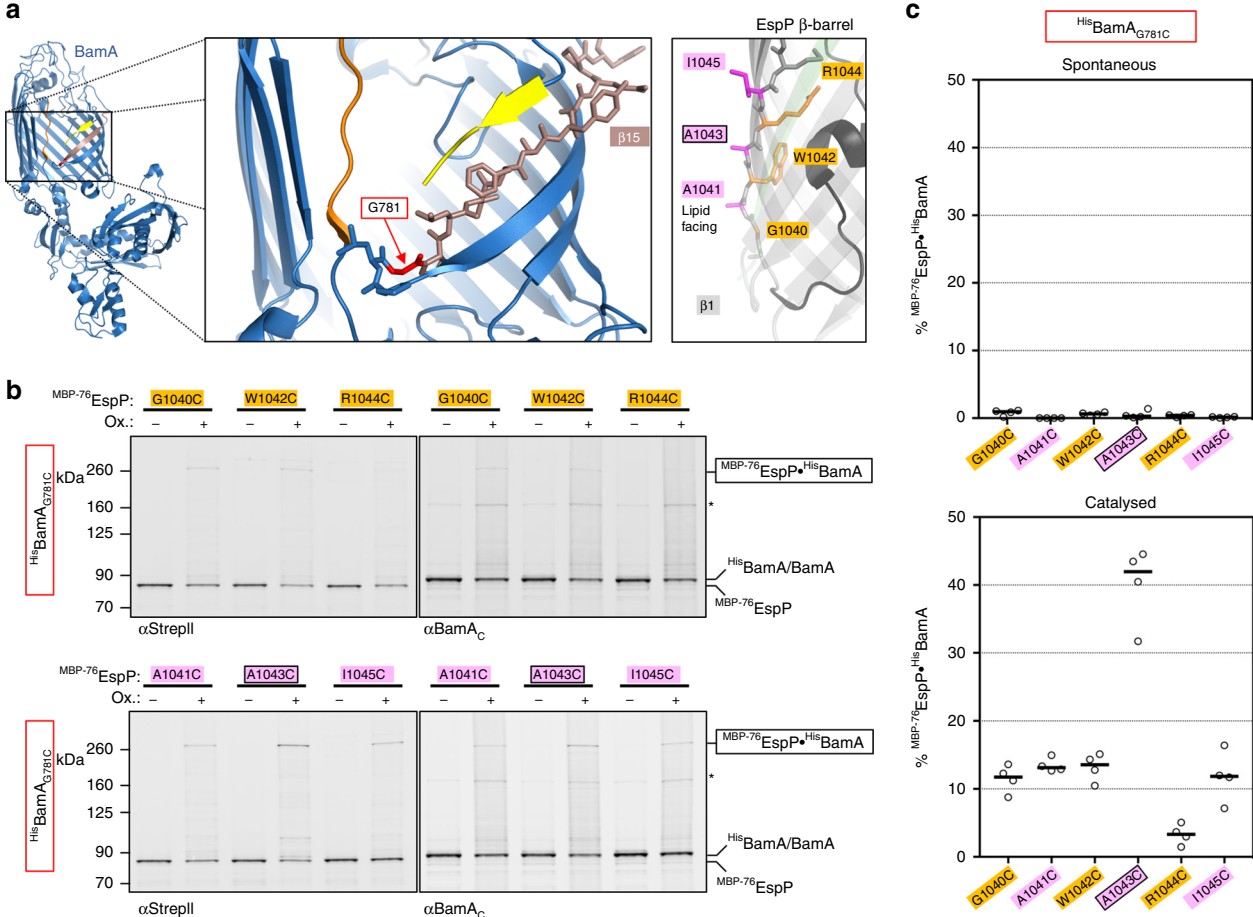

**Fig. 5 Strong bond between a mid-strand $^{MBP-76}$EspP(β1) cysteine and a periplasmic-proximal BamA(β15) cysteine. a** Left, view of periplasmic-proximal BamA β-strand 15 residue G781 (red) in the lateral opening. Right, view of $^{MBP-76}$EspP(β1) showing lipid-facing (magenta) and luminal (orange) residues. **b** BL21(DE3) that expressed $^{MBP-76}$EspP with a single cysteine substitution in β1 and $^{His}$BamABCDE with the $^{His}$BamA$_{G781C}$ substitution were mock-treated (−) or treated with 4-DPS (+). Duplex-immunoblots were then conducted using antibodies/antisera against the N-terminus of $^{MBP-76}$EspP (αStrep II) and the C-terminus of $^{His}$BamA (αBamA$_C$) to monitor disulfide-bond formation between cysteine pairs in vivo. Non-specific bands are denoted (*). Data are representative of at least two independent experiments. **c** Quantitation of disulfide-bond formation between $^{MBP-76}$EspP(β1) and $^{His}$BamA$_{G781C}$ cysteine pairs in mock-treated cells ('spontaneous') and 4-DPS-treated cells ('catalysed'). Experiments were performed as in **b**, except that only αStrepII was used for probing immunoblots. Bars = median, N = 4. ANOVA and multiple comparison tests are shown in Supplementary Table 5. Source data are provided as a Source Data file

heterogeneous interactions between the outward-facing surface of BamA(β15/β16) and $^{MBP-76}$EspP(β1) that were not observed in the analysis of Sam50 function[18].

Because our analysis of F785 interactions strongly suggested that the outward-facing surface of BamA(β15) creates a unique interface with $^{MBP-76}$EspP(β1), we sought to investigate BamA (β15) interactions at positions closer to the periplasmic side of the OM. Perhaps because L783 and P782 are essential for function, we could not introduce cysteine substitutions at these positions. However, we were able to obtain $^{His}$BamA$_{G781C}$ and to assess the formation of disulfide-bonds between this residue and the aforementioned $^{MBP-76}$EspP(β1) cysteine substitutions (Fig. 5a). Upon the addition of oxidiser most cysteine-pairs formed a modest level of disulfide-bonded adducts (~10%) that was similar to the levels observed in the analysis of $^{His}$BamA$_{F785C}$ (Fig. 5b, c, Supplementary Fig. 8). However, the level of disulfide-bond formation between the lipid-facing $^{MBP-76}$EspP$_{A1043C}$ residue and $^{His}$BamA$_{G781C}$ was considerably higher (42%) than that formed between any other $^{MBP-76}$EspP(β1)/$^{His}$BamA(β15/β16) cysteine-pair (Fig. 5b, c). The results suggest that $^{MBP-76}$Esp-P$_{A1043C}$ and $^{His}$BamA$_{G781C}$ are in close proximity in the

most probable of multiple conformational states that can be occupied by $^{MBP-76}$EspP(β1) and the C-terminus of BamA when the assembly of $^{MBP-76}$EspP stalls. Importantly, the interaction of A1043C, which is located near the middle of the OM spanning segment of $^{MBP-76}$EspP(β1), with $^{His}$BamA$_{G781C}$, which is located close to the periplasm, strongly suggests that $^{MBP-76}$EspP(β1) is partially in the periplasm and not stably integrated into the OM in this conformational state.

**The two $^{MBP-76}$EspP-BamA interfaces exhibit distinct features.** Several observations suggested that when the assembly of $^{MBP-76}$EspP is arrested, it forms two very different barrel-barrel interfaces with BamA. The interaction of BamA(β1) and the $^{MBP-76}$EspP β-signal created a rigid, antiparallel, non-sliding inter-barrel β-seam. The formation of disulfide-bonds even in the absence of an oxidant attested to the high stability of this interaction. In contrast, $^{MBP-76}$EspP(β1) did not form an analogous interface with $^{His}$BamA(β16) but rather weaker, more flexible, interactions with outward-facing positions in both $^{His}$BamA(β15) and $^{His}$BamA(β16) that were dependent

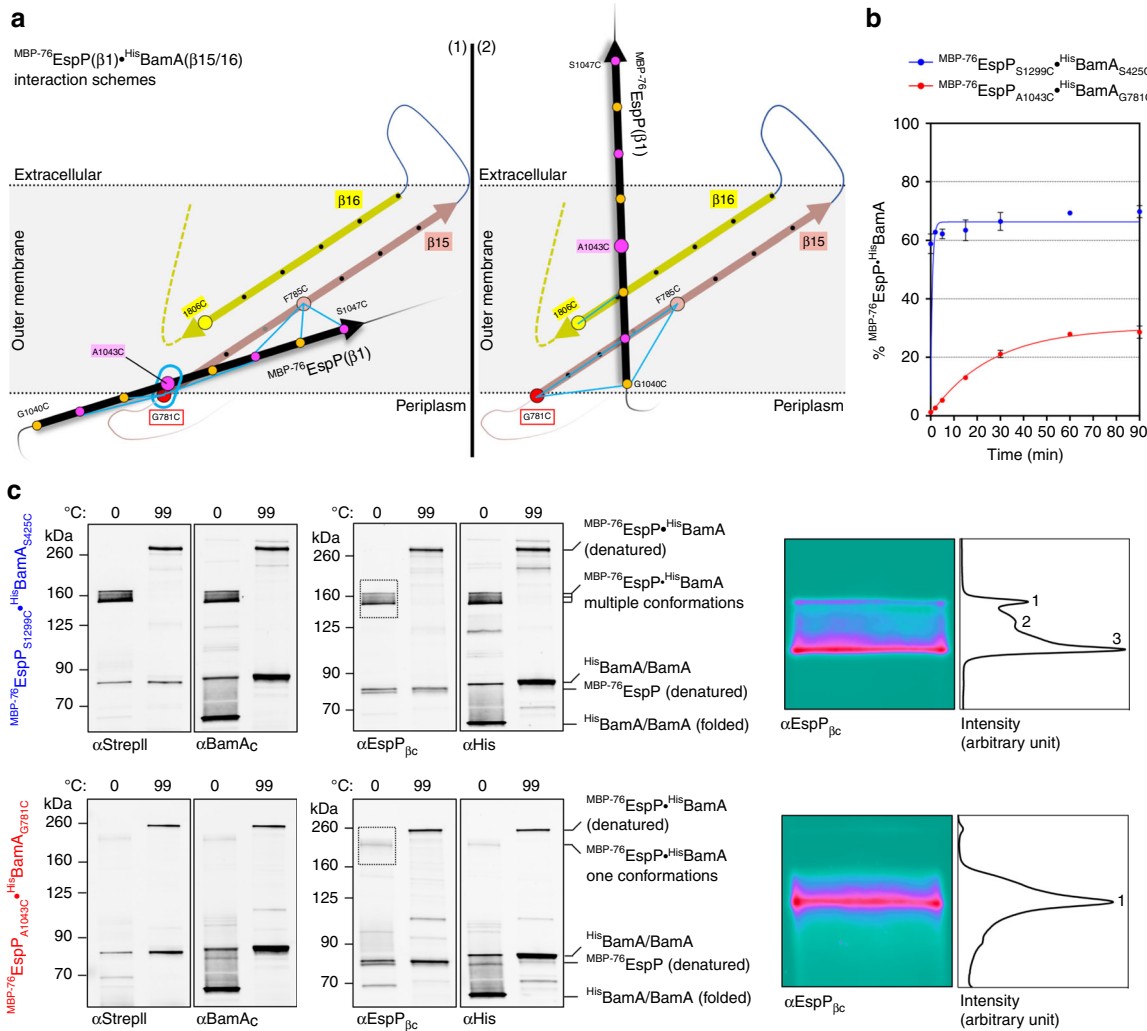

**Fig. 6 Distinct properties of the two interfaces formed between $^{MBP-76}$EspP and BamA. a** Models of interactions between $^{MBP-76}$EspP(β1) and BamA(β15/β16) based on disulfide-bond formation data in Figs. 4 and 5. $^{MBP-76}$EspP(β1) (black), and antiparallel BamA β15 (salmon) and β16 (yellow), are shown as arrows. The BamA β-strands are tilted to match their orientation in solved structures. Key residues are represented by coloured circles. Cysteine-pairs that formed disulfide-bonds at a median level of 10–20% are denoted by blue lines while levels above 40% are denoted by a blue oval. Although no single model can completely explain the observed disulfide-bond patterns, placement of $^{MBP-76}$EspP(β1) in the two orientations that are shown accounts for most of the data. **b** BL21(DE3) that expressed $^{MBP-76}$EspP$_{S1299C}$/$^{His}$BamA$_{S425C}$ (blue) or $^{MBP-76}$EspP$_{A1043C}$/$^{His}$BamA$_{G781C}$ (red) pairs were treated with 4-DPS on ice over a 90 min time course and the kinetics of disulfiide-bond formation was quantitated from immunoblots probed with αStrepII. The $^{MBP-76}$EspP$_{S1299C}$/$^{His}$BamA$_{S425C}$ data could not be fit to a single-step binding curve ($R^2 = -14.94$), while the $^{MBP-76}$EspP$_{A1043C}$/$^{His}$BamA$_{G781C}$ data could be fit ($R^2 = 0.98$, K = 0.038 ± 0.004 min$^{-1}$, half-time = 18.2 min). Points = mean ± standard error, N = 3. **c** BL21(DE3) that expressed $^{MBP-76}$EspP$_{S1299C}$/$^{His}$BamA$_{S425C}$ or $^{MBP-76}$EspP$_{A1043C}$/$^{His}$BamA$_{G781C}$ pairs were treated with 4-DPS. Cell lysates were either unheated (0 °C) or heated (99 °C) and proteins were resolved by cold SDS-PAGE. Duplex-immunoblots were then conducted using αEspP$_{βC}$/αHis or αStrepII/αBamA$_C$ antisera/antibodies. Right, image analysis of the boxed regions of the blots was performed using Fiji software and false colouring. Horizontal average intensity plots (arbitrary values) are shown, and distinct peaks are numbered. Representative results from at least two independent experiments are shown. Source data are provided as a Source Data file

on the addition of an oxidant. Because no single placement of $^{MBP-76}$EspP(β1) relative to $^{His}$BamA(β15/16) can account for all of the disulfide-bonds we observed (Fig. 6a), it is likely that multiple conformations that reflect snapshots of different stages of assembly existed.

To further examine the asymmetry between the two barrel-barrel interfaces, we compared the kinetics of disulfide-bond formation between pairs $^{MBP-76}$EspP$_{S1299C}$/$^{His}$BamA$_{S425C}$ (representing the $^{MBP-76}$EspP(β12-$^{His}$BamA(β1) interface) and $^{MBP-76}$EspP$_{A1043C}$/$^{His}$BamA$_{G781C}$ (representing a major conformer of the $^{MBP-76}$EspP(β1)-$^{His}$BamA (β15/β16) interface). A large fraction of the $^{MBP-76}$EspP$_{S1299C}$/$^{His}$BamA$_{S425C}$ pair formed

adducts extremely rapidly and plateaued almost immediately, while cells that expressed $^{MBP-76}$EspP$_{A1043C}$/$^{His}$BamA$_{G781C}$ displayed a slow accumulation of adducts during an extended incubation period before plateauing at a much lower level (Fig. 6b, Supplementary Fig. 9a, b). The results show a clear difference in the association stabilities between the two inter-barrel interfaces and presumably reflect a difference in local conformational flexibility. The difference in plateaus also provides evidence that when only interactions between $^{His}$BamA$_{G781C}$ and $^{MBP-76}$EspP$_{A1043C}$ are monitored, alternate conformations between $^{MBP-76}$EspP(β1) and the C-terminus of $^{His}$BamA remain unaccounted for.

To obtain direct evidence that the two inter-barrel interfaces have distinct conformational properties, we exploited the remarkable ability of folded β-barrels to resist SDS denaturation in the absence of heat and migrate relatively rapidly on SDS-PAGE. As in the previous experiment, we expressed either $^{MBP-76}EspP_{S1299C}/^{His}BamA_{S425C}$ or $^{MBP-76}EspP_{A1043C}/^{His}BamA_{G781C}$ pairs and catalysed disulfide-bond formation with 4-DPS. Cell lysates were then kept on ice or boiled, and proteins were resolved by SDS-PAGE at 0–4 °C. Interestingly, the unheated $^{MBP-76}EspP_{S1299C}$-$^{His}BamA_{S425C}$ adduct was detected on duplex immunoblots as at least three discrete fast-migrating species that likely represent distinct conformational states (Fig. 6c, top). Identical results were obtained when the experiment was repeated with another $^{MBP-76}EspP(β12)$-$^{His}BamA(β1)$ pair ($^{MBP-76}EspP_{N1293C}/^{His}BamA_{G431C}$) (Supplementary Fig. 9b, c). The unheated $^{MBP-76}EspP_{A1043C}$-$^{His}BamA_{G781C}$- adduct, however, was detected as only a single fast-migrating species (Fig. 6c, bottom). Consistent with the results of previous experiments, these results suggest that when the stable $^{MBP-76}EspP(β12)$-$^{His}BamA(β1)$ inter-barrel β-seam is tethered by a disulfide-bond, the opposing $^{MBP-76}EspP(β1)$-BamA C-terminus interface remains free to adopt multiple intermediate conformations. Conversely, when the $^{MBP-76}EspP(β1)$-$^{His}BamA(β15)$ interface is locked, only one conformation is observed because the opposing inter-barrel β-seam exists as a single stable state.

## Discussion

In this study we describe a structure-guided interaction map of BamA bound to an OMP assembly intermediate in vivo with disulfide-bond resolution. To perform our analysis, we first engineered a fusion protein in which an MBP moiety arrests the translocation of an autotransporter passenger domain and thereby traps the β-barrel in a stable, incompletely assembled state. After validating $^{MBP-76}EspP$ as an assembly intermediate by showing that the arrest of assembly is reversible, we pinpointed positions in BamA and $^{MBP-76}EspP$ β-barrels that are in close proximity by monitoring intermolecular disulfide-bond formation after the addition of an oxidant. We found that the two proteins interact via two distinct interfaces that create an asymmetric hybrid-barrel. On one side, the EspP β-signal formed a rigid antiparallel inter-barrel β-seam with BamA(β1). This observation rules out an 'assisted' model in which BamA catalyses assembly simply by perturbing the lipid bilayer. Unexpectedly, no analogous interface between BamA(β16) and $^{MBP-76}EspP(β1)$ was observed. Instead, we identified diverse, relatively weak interactions between the outward-facing surface of BamA(β15/β16) and $^{MBP-76}EspP(β1)$ that indicated the presence of multiple conformations. Remarkably, in the most common conformer the middle of the $^{MBP-76}EspP(β1)$ transmembrane segment was positioned near the most periplasmic-proximal position of BamA (β15). In this state, part of the EspP β-barrel presumably remained in the periplasm. Finally, an analysis of disulfide-bond formation kinetics and mobility states on SDS-PAGE provided direct evidence that one inter-barrel interface is stable, while the other is conformationally heterogeneous.

Although hybrid-barrel formation and an analogous interaction between Sam50(β1) and the β-signal of client proteins was also recently reported in an examination of interactions between Sam50 and C-terminal fragments of mitochondrial β-barrels[18], our results differ from those of the previous study in several critical respects. First, the previous study reported a potentially strong interface between Sam50(β16) and the N-terminal β-strand of the client that we did not observe in our analysis of the interaction of BamA and $^{MBP-76}EspP$. Second, a predominant conformational state in which the first β-strand of an incoming

β-barrel does not appear to be fully integrated into the OM ($^{MBP-76}EspP_{A1043C}$-$^{His}BamA_{G781C}$) was not identified in the Sam50 study. Most significantly, while we observed disulfide-bond formation between $^{MBP-76}EspP(β1)$ and the outward-facing surface of BamA(β15/16), chemical crosslinking of N-terminal residues of mitochondrial β-barrel fragments to luminal residues of Sam50(β15) and an internal loop of Sam50 was observed in the previous study. Those results provided evidence for a model involving initial threading of unfolded β-barrels into the Sam50 lumen, progressive folding between the two sides of an open Sam50 lateral-gate, and release of the full-length protein into the lipid bilayer[18]. Similar models for OMP assembly that posit the stepwise formation of a hybrid-barrel within the plane of the OM have been proposed based on the crystal structures of BamA[24] and TamA[48], another member of the Omp85 family.

There are several possible explanations for the discrepancies between our study and the Sam50 study. It is conceivable that the use of β-barrel fragments in the Sam50 study captured an earlier stage of the assembly process than we observed through the use of a complete β-barrel. Alternatively, N-terminally truncated mitochondrial β-barrels might interact differently with Sam50 than native clients in vivo. An especially intriguing possibility, particularly in light of the functional diversity of members of the Omp85 superfamily[5,17], is that the catalytic mechanisms of BamA and Sam50 have diverged. Indeed, S. cerevisiae Sam50 and E. coli BamA share only 21% sequence similarity, have a different number of POTRA domains, function in dissimilar membrane environments, and form complexes with unrelated accessory proteins that have opposite membrane topologies[11]. Furthermore, Sam50 and BamA may catalyse the assembly of client proteins that have distinct folding requirements. For instance, bacterial OMPs are profoundly structurally diverse and contain an even number of β-strands that necessitates the formation of an antiparallel β-seam, while mitochondrial β-barrels (besides Sam50) are all members of a single family of 19-stranded proteins that have a parallel β-seam[11,49]. Importantly, if the assembly mechanisms used by Sam50 and BamA have diverged, it might be possible to design antibiotics that target BamA[50] but that do not cause mitochondrial toxicity.

Regardless of the reason that disparate results were obtained in the two studies, we propose that BamA functions by actively integrating client proteins into the OM from the periplasm via a 'swing' mechanism rather than progressively threading them through a lateral gate. In this model, the β-signal of a partially folded client protein forms a transiently stable interaction with BamA(β1) that creates an inter-barrel β-seam (Fig. 7, stage i; Supplementary Video 1). The inter-barrel β-seam maintains the association between the two barrels while BamA acts as a molecular hinge that allows the N-terminus of the client β-barrel to move along the outward-facing surface of its C-terminal strands and integrate into the OM through a swinging action (Fig. 7, stage ii). Although we cannot order a series of intermediate conformations from our steady-state data, it seems likely that the diverse interactions that we observed between $^{MBP-76}EspP(β1)$ and lipid-exposed residues of BamA(β15/β16) represent assembly snapshots of the partially folded OMP moving along the outward-facing surface of BamA into the plane of the OM. The strong $^{MBP-76}EspP_{A1043C}$-$^{His}BamA_{G781C}$ interaction (which in itself provides clear evidence that the β-strands do not simply partition into the lipid bilayer through a lateral-gate) presumably corresponds to an early state in which much of the client is still positioned in the periplasm. In a final step, the client is released from BamA into the lipid bilayer as a fully assembled β-barrel (Fig. 7, stage iii). We speculate that the asymmetry helps complete assembly by promoting the energetically favourable closure of the

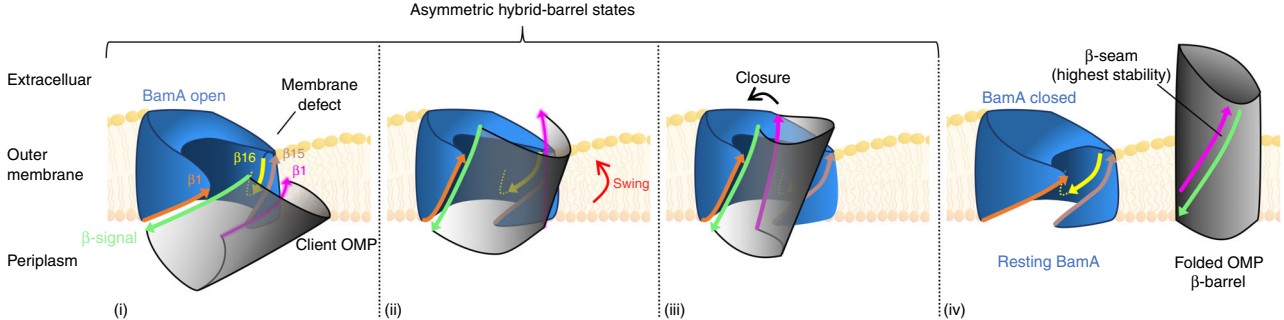

**Fig. 7** Model of bacterial outer membrane protein assembly. Red arrows indicate possible conformational changes. After an OMP (grey) is targeted to the OM, it interacts with the BamA β-barrel (blue). Initially the C-terminal β-strand of the OMP (β-signal, green) forms a β-seam with BamA β1 (orange) to create an asymmetric hybrid barrel (i). It is not yet clear if the OMP begins to fold before or after the formation of this interface. Subsequently the first β-strand of the OMP (magenta) forms a low stability interface with the outward-facing surface of BamA β15/16 (salmon/yellow) (ii). A large-scale movement of the OMP into the membrane ('swing') is aided by the rotation of BamA β1-β8, the perturbation of the lipid bilayer by the wedge-shape of the BamA β-barrel, and the dynamicity of β16. The asymmetry between the two interfaces favours the closure of the client β-barrel (iii) and its release into the lipid bilayer in a native conformation (iv). Our results suggest that a loop of the $^{MBP-76}$EspP passenger domain is exposed on the cell surface prior to most of the swinging action

client β-barrel and restoration of the BamA ground state through a β-signal exchange (Fig. 7, stage iv).

The results of numerous previous studies on OMP biogenesis are consistent with a 'swing' model for BamA function. First and foremost, there is now considerable evidence from in vivo studies on EspP, trimeric adhesins and the LPS transporter complex LptDE that OMPs begin to fold prior to their integration into the OM[27–32] and are therefore unlikely to be inserted by progressive threading. Many of the results suggest that folding begins in the periplasm or is promoted by interactions with the BAM lipoproteins or the BamA POTRA domains that precede interactions with the BamA β-barrel. As suggested in a recently proposed 'elongation' model, however, it is conceivable that the initial association of the β-signal with BamA(β1) nucleates subsequent C- to N-terminal folding on the periplasmic side of the OM[10]. Second, evidence that the effect of lipid-facing arginine residues on OMP biogenesis is highly dependent on membrane depth is consistent with the notion that charged groups hinder productive passage through a hydrophobic environment[51]. Structural studies also support a 'swing' model. Solved structures of BAM show that the transition between the 'closed' and 'open' states of the BamA β-barrel involves a remarkable scissor-like 45°–65° rotation of BamA(β1–β8)[19,20] that could represent the hinge mechanism for a major swinging action of client OMPs. There is also substantial evidence that the wedge-shaped aromatic-girdle of BamA and the highly dynamic properties of BamA(β16) promote membrane thinning and disorder near the lateral opening[24–26,47]. This membrane perturbation would potentially drive the swinging movement of partially folded β-barrels into the OM by lowering the energy barrier for integration and by promoting hydrophobic interactions between their outer surface and membrane lipids. Finally, our model fits well with recent bioinformatic evidence that the C-terminal half of OMPs is especially well conserved[52]. We speculate that the need to form an asymmetric hybrid-barrel during assembly constrains the C-terminal sequences of OMPs but provides the freedom for evolution through N-terminal modifications[52].

## Methods

**Plasmid construction.** The plasmids that were used in our experiments are listed in Supplementary Table 6. Oligonucleotide primers and dsDNA fragments that were used in plasmid construction are listed in Supplementary Table 7. To construct pMTD321, a plasmid that encodes an MBP fusion to the EspP β-barrel with a 115aa-linker, a DNA fragment encoding *malE* was PCR amplified using primers Mal.Eag(+)/Mal.Eag(−) and pMAL-p2x as a template, digested with EagI, and

ligated into the EagI site of pJH64[53] to create pJH64::*malE*. The open reading frame of the fusion protein was then PCR amplified using primers mtd1/pTRC 399 R and pJH64::*malE* as a template, digested with NdeI and HindIII, and sub-cloned into similarly digested pWK1[38], a derivative of the rhamnose-inducible plasmid pSCrhaB2[43]. Finally, the resulting plasmid was PCR amplified with primers mtd96/mtd97 and mixed with dsDNA fragment mtd95 to attach a TwinStrepII-tag to the fusion protein via Gibson assembly. To construct plasmids that encode MBP fusions to the EspP β-barrel with 97, 76, or 59aa-linkers (pMTD501, pMTD502 and pMTD610, respectively), pMTD321 was PCR amplified using primers mtd145/mtd146, mtd145/mtd147, or mtd145/164, respectively, digested with BamHI, and re-circularised. To make pMTD521 and pMTD607 (pRha$^{MBP-76}$EspP), which encode an MBP fusion to the EspP β-barrel with a 76aa-linker and a TEV cleavage site between EspP residues 974/975 or 984/985, respectively, pMTD502 was PCR amplified using primers mtd135/136 or mtd160/161 and assembled with mtd137 or mtd159 dsDNA fragments (encoding TEV cleavage sites). Plasmid pMTD826, which encodes $^{76}$EspP, was generated by first introducing a second BamHI site into pMTD607 using primers mtd186/187 together with the QuikChange II Site-Directed Mutagenesis Kit (Agilent). The resulting plasmid (pMTD798) was then digested with BamHI and re-circularised. To construct pMTD372 (pTrc99a::$^{His8}$bamABCDE), pJH114[34] was PCR amplified using primers mtd102/mtd103 and assembled with dsDNA fragment mtd101 (encoding a His8 tag). The resulting plasmid (pMTD366) was subsequently mutagenised using primers mtd104/mtd105 to re-introduce the native *bamE* stop codon. Cysteine substitution mutations were introduced into pMTD372 and pMTD607 by site-directed mutagenesis as described above.

**Culture conditions.** *E. coli* B strain BL21(DE3) (Invitrogen catalog number C600003) was used in all experiments. Cells transformed with appropriate plasmids were grown from a single colony in Lysogeny broth (LB) (Miller) at 25 °C with orbital shaking at 250 rpm overnight. Overnight cultures were pelleted (3000 × *g*, 5 min, 4 °C), washed, and resuspended with one culture volume of LB before seeding sub-cultures at $OD_{600} = 0.05$ in Erlenmeyer flasks with loose lids. Subcultures were grown for 4 h (25 °C, 250 rpm) to $OD_{600} \sim 0.4$–0.6, induced with 0.4 mM IPTG for 1 h, then induced for 45 min with 0.2% L-rhamnose. Growth media were supplemented with ampicillin (100 μg mL$^{-1}$) or trimethoprim (50 μg mL$^{-1}$) as necessary.

**Western immunoblotting, imaging and quantitation.** Proteins were separated by SDS-PAGE on 8–16% Tris-glycine gels (Invitrogen) and transferred to nitro-cellulose membranes using an iBlotII (Life Technologies). Immunoblot buffer [Odyssey Blocking Buffer (Licor) diluted by half with PBS and supplemented with 0.01% Tween20] was used for all blocking and antibody/antisera incubations, PBS + 0.01% Tween20 (PBST) for initial washes, and PBS for final washes. Monoclonal mouse anti-StrepII and anti-His antibodies were obtained from Qiagen (catalog number 34850) and Genscript (catalog number A00186), respectively. Polyclonal rabbit antisera raised against EspP β-barrel and BamA C-terminal peptides have been described previously[30]. A polyclonal rabbit antiserum was also generated against a peptide derived from the N terminus of the EspP β-barrel domain (NH$_2$-NLNKRMGDLRDINGEAGAWARC-COOH). Secondary antibodies [goat anti-mouse 800CW (IRDye catalog number 926–32210) and goat anti-rabbit 680LT (IRDye catalog number 926–68021)] were obtained from Licor. Membranes for quantitation were blocked for 1 h, incubated with anti-StrepII (1:5000 dilution) for 1 h, washed (3 × 5 min with PBST), incubated with goat anti-mouse 800CW (1:5000 dilution) for 1 h, washed (3 × 5 min with PBST, 2 × 5 min PBS), and air

dried. The membranes were then scanned using an Amersham Typhoon 5 imager (GE Healthcare) with a 785 nm laser, IRlong 825BP30 filter, and PMT set at 700 V. Pixel intensities of detected proteins were measured using Fiji software (v2.0.0-rc-68/1.52 g). Within-lane values were used to calculate percent disulfide-bond formation $[(^{MBP-76}EspP-^{His}BamA/(free \ ^{MBP-76}EspP + ^{MBP-76}EspP-^{His}BamA) \times 100]$. Otherwise, membranes for single- or duplex-immunoblots were typically incubated overnight with primary antibodies/antisera and for 2 h with secondary antibodies. Dried membranes were imaged with an Odyssey infrared imager (Licor, model 9120) or a Typhoon 5 imager using maximum quality and resolution settings. Uncropped images of all blots are included in the Source Data file.

**Cell surface protease digestions and arrest-release assembly**. To monitor the surface exposure of the $^{MBP-76}EspP$ linker or the protease sensitivity of BamA loop 6, 1 mL samples of induced bacterial subcultures were aliquoted into 1.5 mL tubes on ice. Cells were pelleted ($10,000 \times g$, 2 min, 4 °C), resuspended in 0.5 mL PBS, and incubated on ice for 20 min (unless otherwise indicated) with 200 µg mL$^{-1}$ Proteinase K (PK) or equivalent volume of PK buffer (5 mM CaCl$_2$, 50 mM Tris-HCL pH 8) for the mock-treated control. For TEV digestions, cells were resuspended in M9 medium with 2 mM DTT and incubated at 25 °C for 2 h with 50 µg mL$^{-1}$ TEV protease (hyperactive S219V mutant, obtained from Dr. Susan Buchanan). To observe the completion of β-barrel assembly following the release of MBP, cells were resuspended in PBS at 25 °C and incubated with 200 µg mL$^{-1}$ PK [25 °C, 350 rpm, in a Thermomixer (Eppendorf)] for 0.5, 2, 10 and 30 min. Cells were then pelleted ($10,000 \times g$, 20 s, 4 °C), resuspended in 0.5 mL PBS, and incubated with 4 mM PMSF and 10% (v/v) TCA on ice for 10 min to inhibit PK and precipitate proteins. TCA precipitates were pelleted ($20,817 \times g$, 10 min, 4 °C), washed with 0.6 mL acetone, pelleted again, and air dried at 37 °C for 15 min. Dried precipitates were resuspended with 2x SDS protein gel loading solution (Quality Biological) in a volume normalised to the final OD$_{600}$ reading (volume, µL = OD$_{600} \times 200$) and heated at 99 °C for 15 min.

**Disulfide-bond formation assay**. To observe site-specific intermolecular protein-protein interactions, 1 mL samples of induced subculture cells were aliquoted into 1.5 mL tubes on ice, pelleted ($10,000 \times g$, 2 min, 4 °C), and resuspended in 1 mL PBS. Cells were incubated with the thiol-specific oxidiser 4,4′-dipyridyl disulfide (4-DPS) at a final concentration of 0.2 mM (or an equivalent volume of ethanol for mock-treated controls) and incubated on ice for 30 min. Cells were then pelleted ($10,000 \times g$, 2 min, 4 °C), resuspended in 0.5 mL PBS, and mixed with TCA to precipitate proteins as described above. To monitor the kinetics of intermolecular disulfide-bond formation, 5 mL samples of induced subculture were aliquoted into 50 mL tubes on ice, pelleted ($3000 \times g$, 4 min, 4 °C), and washed with 10 mL PBS. Cells were then resuspended in 5 mL PBS and incubated with 0.2 mM 4-DPS or ethanol (mock-treated) for 0, 2, 5, 15, 30, 60 and 90 min. At each time point 0.4 mL aliquots were dispensed into 1.5 ml tubes pre-loaded with PMSF and TCA for instant protein precipitation. TCA precipitates were washed and mixed with 2x SDS protein gel loading solution as described above. Control samples were reduced with 150 mM dithiothreitol (DTT) during heating.

**Release of assembly arrest and disulfide-bond reduction**. To restart β-barrel assembly following the release of MBP and the reduction of $^{MBP-76}EspP-^{His}BamA$ inter-barrel disulfide-bonds, 20 mL samples of induced subculture were aliquoted into 50 mL tubes on ice and pelleted ($3000 \times g$, 5 min, 4 °C). Cells were washed with 40 mL PBS, pelleted ($4000 \times g$, 6 min, 4 °C), and resuspended in 10 mL PBS. Cell aliquots (0.5 mL) were placed into 1.5 mL tubes, incubated with 0.2 mM 4-DPS for 30 min on ice, pelleted ($10,000 \times g$, 2 min, 4 °C), and resuspended in 0.5 mL PBS. Cells were then incubated with either 200 µg mL$^{-1}$ PK or PK buffer (mock digest) for 30 min on ice, washed with 1 mL PBS, and resuspended in 0.5 mL PBS (containing 1 mM PMSF) pre-warmed to 25 °C. Cells were then incubated in a Thermomixer (25 °C, 350 rpm) either in 50 mM HEPES pH 7 containing 150 mM DTT or without DTT (mock-treated) for 0.5, 2, 5 and 15 min. All samples were TCA precipitated on ice and prepared for SDS-PAGE as above.

**Gel mobility-shift assay**. To observe hybrid-barrel conformational states, cysteine pairs were oxidised as described above. Cells were then resuspended in BugBuster Master Mix (EMD Millipore) containing EDTA-free SigmaFast protease inhibitor (Sigma-Aldrich) (volume, µL = OD$_{600} \times 100$) and lysed on ice for 3 min. Aliquots (30 µL) of lysates were mixed with 10 µL 2x SDS protein gel loading solution to bring the final SDS concentration to 1%. Samples were either maintained on ice or heated at 99 °C for 10 min. Proteins were then resolved by 'cold' SDS-PAGE (i.e., by packing gel tanks in ice and running the gels in a 4 °C room) and transferred to nitrocellulose for immunoblotting as described above.

## Data availability

Data supporting the findings of this paper are available from the corresponding author upon reasonable request. A reporting summary for this Article is available as a Supplementary Information file. The source data underlying Figs. 1d–e, 2b–e, 3b–c, 4b–e, 5b–c, 6b–c and Supplementary Figs. 1a–c, 2a–c, 3a–d, 4a–c, 5, 6a–e, 7a–e, 8, 9a–d are provided as a Source Data file.

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

## Acknowledgements

We would like to thank Janine Peterson for technical assistance in the subcloning of *malE* and Sunyia Hussain for critical reading of the paper. This work was supported by the Intramural Research Program of the National Institute of Diabetes and Digestive and Kidney Diseases.

## Author contributions

The study was conceived and designed by M.T.D. and H.D.B. The experimental work was conducted by M.T.D. The paper was written by M.T.D. and H.D.B.

## Additional information

**Competing interests:** The authors declare no competing interests.

