## [Peer Review File · Nature Communications]

Reviewers' Comments:

Reviewer #1:

Remarks to the Author:

Review of "Bacterial outer membrane proteins assemble via asymmetric interactions with the BamA β -barrel" by Doyle and Bernstein

The β -barrel assembly machine (BAM) folds and inserts β -barrel proteins into the outer membrane, an essential process in Gram-negative bacteria. A related complex carries out the same function in mitochondria and chloroplasts. Despite extensive genetic, biochemical, and structural data, how β -barrel protein substrates interact with BAM remains unknown. Doyle and Bernstein have developed an approach to capture a β -barrel BAM substrate in the process of being inserted in cells. They dissected this intermediate assembly complex by analyzing the ability of introduced cysteines to form intermolecular disulfide bonds. Their results suggest a new model, termed the swing mechanism, to explain how BAM inserts β -barrel proteins into the outer membrane.

The paper is well-written, the presented data are clear and novel, and the authors' interpretations are logical and justified. These findings provide interesting and convincing new data that will allow for a greater understanding of the fundamental process of β -barrel membrane protein folding. This manuscript will be of high interest to the broad audience of Nature Communications.

While not critical, below are a couple minor points/questions the authors can consider.

1. The swing model is compelling and fits with available data. Given other available information in the field, there are a couple steps that could be flushed out a little more when describing this model:

- Do the POTRA domains play any role in the swing step, or are they likely involved in a different step in OMP folding?
- What about the role of the BAM lipoproteins, especially the essential lipoprotein BamD? (Likely for future experiments: the authors' approach could be used in strains with *bamB*, *bamC*, or *bamE* deletions or with *bamA* mutants known to disrupt BAM complex formation)
- What is the driving force for the swinging movement that ultimately moves the pre-folded β -barrel across the outer membrane (i.e., what triggers the β -barrel to 'swing' and how is that powered)?
- In the final step when the β -barrel substrate must be separated from BamA, what 'disrupts' the stable interaction between BamA(β 1) and EspP(β 12)? Is the interaction between EspP(β 1)-EspP(β 12) (or possibly between BamA(β 1)-BamA(β 16)) stronger than the BamA(β 1)-EspP(β 12) interaction? Or is the orientation/geometry of the interactions altered post-swing?
- Extracellular loop 6 has been suggested to play an important role in BamA function (and in Sam50 function as well). Can this be accounted for in the model? (Also, for likely future experiments: the authors' approach can be applied to loop 6 mutants)

2. In the previous dissection of SAM, simultaneous interactions were observed between both sides of the Sam50 β -seam and two β -strands of the substrate (β 19 (the β -signal) and β 14). Presumably, such an interaction with an internal β -strand of the substrate would not be needed (or found) for the swing model with BAM. Did the authors consider looking for interactions (or lack of interactions) between BamA β 15/16 and any of the other EspP β strands (2-11)? Or simultaneous interactions with both sides of the seam? One other interaction that should be lacking in the swing model that was not examined is BamA(β 15/16) with EspP(β 12). From what I could tell, all the other important potential interactions seem to have been examined: BamA(β 1)-EspP(β 12), BamA(β 2)-EspP(β 12), BamA(β 1)-EspP(β 1), BamA(β 15/16)-EspP(β 1).

3. Regarding the 'sliding' interpretation of the multiple interactions observed between EspP(β 1) along BamA(β 15/16): While I agree this is a reasonable interpretation, could an alternative be that these are not different steps in the process, but rather that there are just multiple solutions to the BamA(β 15) and EspP(β 1) interactions, any one of which could be productive and facilitate the swing? Also, if there is sliding, one could imagine that there is a potential for EspP(β 1) to interact with positions closer to the cell surface (e.g. BamA 789 or 801 or some other feature). The authors teased apart the interactions proximal to the periplasm, but, unless I missed it, there is still some space to explore potential interactions closer to the cell surface.

4. A couple other minor points:

- Line 111. Might be convenient to refer to Figure 1b at the end of this sentence.
- Consider re-numbering the panels in Figure 2 (move the current 2E to 2C)
- Line 232. Is there a possibility that an interface that is not apparent from the available structures (which are solved in the absence of substrate) could be present when substrate is engaged and that was missed? Are there any other positions to consider for interaction?
- Line 379. Although not the simplest explanation, one could also imagine that there are different intermediates for different OMPs (e.g., autotransporters versus OMPs with lipoproteins bound in lumen versus trimeric OMPs versus simple OMPs, etc.). This is the first and only example of an intermediate being trapped on BAM so it will be interesting to see, in the future, how different substrates interact with this complex.

Reviewer #2:

Remarks to the Author:

The manuscript by Doyle and Bernstein describes the structural mapping of a blocked intermediate of the insertase protein BamA inserting a suitably modified version of the substrate protein EspP. Spatial proximities in the blocked intermediate state are identified by chemical cross-linking via engineered disulfide bridges and analyzed by Western blots. They authors find a well-defined interface between strands b1 of BamA and b12 of the substrate and a less-well defined interface with apparently multiple conformations between b1 of EspP and b15/b16 of BamA. The work is very well done, the data are of outstanding quality and everything is well documented and described, along with sufficient control experiments. The demonstration of the existence of a stable intermediate of BamA-EspP and the derivation of some structural information for this protein complex is a significant contribution to understanding Bam and its mechanism. This is a nice piece of work and I clearly support publication in Nat. Comm.

The following points should however be addressed before publication.

- 1.) The present work was done with the autotransporter EspP as a client and extrapolates from these data to all OMPs in general. The model shown in Figure 7 should therefore also include the situation of autotransporter clients and indicate at which point the passenger domains is expected to translocate to the extracellular side. I assume this must be at the hybrid barrel state (ii).
- 2.) The present work investigates the BamA-mediated insertion mechanism of an autotransporter. There has been a previous mechanistic model for autotransporter insertion by the insertase TamA, which is homologous to BamA (Gruss et al, NSMB 20, 1318 (2013)). The two resulting models are very similar: Steps ii-iv are essentially identical, they only differ in the question whether the barrel is pre-formed in the periplasm or not. The previous work needs to be credited and the authors should briefly compare the two mechanisms in the discussion.
- 3.) In the model at stage (i), the client OMP is substantially curved, such that the hydrophobic side of the beta-sheet is concave. Such a shape does not really fit to the topology of the native membrane in three dimensions. It also requires that strand b1 is located somewhere in the membrane center, with a row of unsaturated hydrogen bond-donors and acceptors facing the hydrophobic phase of the bilayer. That is at least how it seems to be depicted in Figure 7. Such a conformation is energetically very unfavorable. The authors should explain in the text how exactly this is possible and/or improve the model stage (i) to be more realistic.
- 4.) In the title and elsewhere, the authors refer to the transition complex as "asymmetric". With this, the authors want to express that the interfaces b1/b12 and b1/b15/b16 are structurally not equivalent. The chosen wording may cause confusion, since a heterodimeric protein complex is a priori always asymmetric. Perhaps a different wording can be found.

Reviewer #3:

Remarks to the Author:

Summary. The biogenesis of beta-barrel outer membrane proteins (OMPs) is still poorly understood, despite the structures of the individual components of BAM being reported, along with recent X-ray and cryoEM structures of fully assembled BAM. It is thought that BAM modulates the membrane bilayer in order to allow the spontaneous or orchestrated insertion of OMPs. Exactly how the OMPs are able to make their way to BAM and the for biogenesis are still elusive. The

authors here use EspP in a crosslinking study of a stalled intermediate to decipher how BAM interacts with EspP during folding into the OM.

Comments (in no particular order):

1. Overall, the paper is well written and organized and presents studies on the biogenesis of OMPs by BAM. The crosslinking methods are a strength of the paper and provide a wealth of information about the interaction of EspP with BAM.
2. The author refer to the interactions as 'asymmetric' interactions, even in the title, yet it is a bit unclear what this refers to as opposed to a 'symmetric' interaction? Could the author elaborate on this, as none of the barrels are symmetric not would one expect a 'symmetric' interaction given this. Possible remove such jargon for clarity?
3. Line 26 – 'open' form of BamA, is this the barrel open to the surface or the POTRA open? Or just that the lateral gate is open? For an interaction to occur with the exposed edge of beta1 of BamA, it would be assumed that the it is a lateral gate in open state.
4. Line 27 – why 'Unexpectedly?'. This study is the first to interrogate such a system, and the results are all new. Maybe it was meant in contrast to the Sam50 study? And if so, it is still not surprising that there are differences between the two systems. Further, this study is based on much more convincing structures than the Sam50 study and therefore easier to interpret.
5. Line 85 – 'asymmetric hybrid-barrel', again, why is asymmetric needed here?
6. Lines 86-87 – maybe it was missed, but no data seemed to be presented either in favor or against a 'threading' model, as only the N-term and C-term strand were studied primarily and assuming these possibly interact first, one would have to interrogate within the client OMP before concluding anything about a 'threading' mechanism or not.
7. EspP is an autotransporter and therefore likely has unique properties and biogenesis from other OMPs. While this doesn't deter from these nice studies, the authors should make this a clear point since the barrel is mostly assembled and just awaiting maturation upon cleavage. This is not like other OMPs that don't have a passenger domain or cleavage activity. Therefore, it is likely that BAM interactions with these class a bit differently in order to allow the passenger biogenesis.
8. Line 156 and Fig 1d/e – the levels in panel d appear very faint however are much more obvious in panel e, what was the different here?
9. Line 176 – 'fold completely' suggests that BamA is not folded properly and possibly dysfunctional. However, the lack of the disulfide is more likely just making the loop more dynamic and therefore more accessible to PK, which is much different than having a mis-folded BamA. Authors should consider revising this for clarity, esp for a more general audience. With the disulfide, the loop is more compact, less dynamic, and less accessible for PK digestion.
10. Throughout the manuscript, the authors refer to the 'outer surface' but this is a bit confusing since it could mean towards the surface side of the bacteria, or the outside of the barrel. Authors should consider making sure this is clear what is meant throughout each portion of the manuscript.
11. Page 16, the authors make the claim that differing conformational states are observed for one crosslink while not for the other. While the gel does appear to show smearing or laddering, one should be cautious of pre-boil samples as this can also be a function of stability also. For example, one may form a tight interaction while the other forms a more loose interaction that slow breaks down as the gel is ran. If these are truly different conformations, PK digests might be able to tease this out and be a nice complement here.
12. The crosslinking studies are nicely done and the authors make a very convincing case for the interaction of bamA beta-1 and espP beta-12, and the same for the beta-1 to some degree. However, the resulting proposed 'hinge' mechanism is much less convincing as noted below:
 - a. The authors are weighing the model based almost entirely on the observations of the beta1 interaction of EspP with BamA, but this is likely the final before insertion by strand exchange. The authors don't appear to consider the initiating step here of beta-12 which seems to be the most stable and wouldn't appear to have the nearly parallel to the membrane conformation as suggested in the model.
 - b. The suggestion that one face of the client OMP faces the periplasm and then inverts itself and concavity seems highly unlikely. While the hydrophilic face would be facing the periplasm, the hydrophobic face would be completely exposed to the surface of the bacteria or the inner side of the membrane, both of which are not likely. Further, what is the force or mechanism by which this conversion would be regulated or mediated, as the protein(s) would not just adopt this transition automatically?
 - c. What is the role of the other Bam components? This mechanism is BamA centric, what would the other Bam proteins be doing here to mediate OMP biogenesis?
 - d. The model depends on a rigid motion of BamA rotating the entire client OMP into the membrane, which would require quite a bit of torque and energy, where would this come from?

e. Overall, this reviewer is less convinced by the proposed model despite being excited about the rest of the manuscript. One could just as convincingly argue that autotransporters undergo a mechanism sharing features of both major models and the data would align equally as well as for the new proposal model. The current model isn't fully supported by the literature or by the studies presented and only raises more questions than piecing things together; therefore, suggest removing the model to strengthen the overall manuscript.

REPLY TO REVIEWERS' COMMENTS

Reviewer #1:

The β -barrel assembly machine (BAM) folds and inserts β -barrel proteins into the outer membrane, an essential process in Gram-negative bacteria. A related complex carries out the same function in mitochondria and chloroplasts. Despite extensive genetic, biochemical, and structural data, how β -barrel protein substrates interact with BAM remains unknown. Doyle and Bernstein have developed an approach to capture a β -barrel BAM substrate in the process of being inserted in cells. They dissected this intermediate assembly complex by analyzing the ability of introduced cysteines to form intermolecular disulfide bonds. Their results suggest a new model, termed the swing mechanism, to explain how BAM inserts β -barrel proteins into the outer membrane.

The paper is well-written, the presented data are clear and novel, and the authors' interpretations are logical and justified. These findings provide interesting and convincing new data that will allow for a greater understanding of the fundamental process of β -barrel membrane protein folding. This manuscript will be of high interest to the broad audience of Nature Communications.

Thanks for the positive feedback!

While not critical, below are a couple minor points/questions the authors can consider.

1. The swing model is compelling and fits with available data. Given other available information in the field, there are a couple steps that could be flushed out a little more when describing this model:

- Do the POTRA domains play any role in the swing step, or are they likely involved in a different step in OMP folding?
- What about the role of the BAM lipoproteins, especially the essential lipoprotein BamD? (Likely for future experiments: the authors' approach could be used in strains with *bamB*, *bamC*, or *bamE* deletions or with *bamA* mutants known to disrupt BAM complex formation)

Our experiments really do not examine the role of the BamA POTRA domains or the BAM lipoproteins in the swing step, so any additional comments would be pure speculation. The structure of BAM shows that the lipoproteins and POTRA domains form a ring-like structure on the periplasmic side of the BamA β -barrel that has been proposed to be a site of substrate recognition and/or initial folding. We state on lines 412-413 that interactions between client proteins and the BAM lipoproteins/BamA POTRA domains may promote folding, and we have now modified the text to indicate that these interactions likely precede the interaction between client proteins and the BamA β -barrel. We agree that it will be interesting to examine the fate of MBP-76^{EspP} in strains containing mutant forms of BAM in the future investigations.

- What is the driving force for the swinging movement that ultimately moves the pre-folded β -barrel across the outer membrane (i.e., what triggers the β -barrel to 'swing' and how is that powered)?

That's a great question—really the million-dollar question in the field. Near the very end of the Discussion we cite evidence that the dynamics of the BamA β -barrel promotes membrane thinning and disorder near the lateral opening and suggest that this membrane perturbation might facilitate the swinging movement by lowering the energy barrier for integration. To expand

on this point, we have now modified the text to suggest that perturbation of the membrane might also promote the formation of hydrophobic interactions between partially folded β -barrel proteins and membrane lipids (lines 426-427).

- In the final step when the β -barrel substrate must be separated from BamA, what 'disrupts' the stable interaction between BamA(β 1) and EspP(β 12)? Is the interaction between EspP(β 1)-EspP(β 12) (or possibly between BamA(β 1)-BamA(β 16)) stronger than the BamA(β 1)-EspP(β 12) interaction? Or is the orientation/geometry of the interactions altered post-swing?

Both hypotheses seem reasonable, although as indicated in Fig. 7 (step iv) and in the text (lines 403-405), we favor the idea that the EspP(β 1)-EspP(β 12) interaction is stronger than the BamA(β 1)-EspP(β 12) interaction.

- Extracellular loop 6 has been suggested to play an important role in BamA function (and in Sam50 function as well). Can this be accounted for in the model? (Also, for likely future experiments: the authors' approach can be applied to loop 6 mutants)

Our results did not generate any information about the function of BamA loop 6. In the interest of basing our model on our data and staying within the length limit, we would rather not speculate on the function of this poorly understood segment of BamA.

2. In the previous dissection of SAM, simultaneous interactions were observed between both sides of the Sam50 β -seam and two β -strands of the substrate (β 19 (the β -signal) and β 14). Presumably, such an interaction with an internal β -strand of the substrate would not be needed (or found) for the swing model with BAM. Did the authors consider looking for interactions (or lack of interactions) between BamA β 15/16 and any of the other EspP β strands (2-11)? Or simultaneous interactions with both sides of the seam? One other interaction that should be lacking in the swing model that was not examined is BamA(β 15/16) with EspP(β 12). From what I could tell, all the other important potential interactions seem to have been examined: BamA(β 1)-EspP(β 12), BamA(β 2)-EspP(β 12), BamA(β 1)-EspP(β 1), BamA(β 15/16)-EspP(β 1).

The reviewer raises a good point in suggesting that we test for interactions between EspP(β 12) and BamA(β 15/16) for the sake of completeness. To address this concern, we have now performed additional disulfide bond formation assays using residue pairs ^{MBP-76}EspP_{N1295C}-^{His}BamA_{N805C}, ^{MBP-76}EspP_{N1295C}-^{His}BamA_{I806C}, ^{MBP-76}EspP_{N1295C}-^{His}BamA_{F785C}, and ^{MBP-76}EspP_{S1299C}-^{His}BamA_{G781C}. The results are shown in a new Figure (Supplementary Fig. 5) and referenced on lines 198-199 of the text. Consistent with the swing model, we did not observe significant disulfide bond formation between EspP(β 12) and BamA(β 15/16).

We did consider looking for interactions between BamA β 15/16 and EspP β strands 2-11, but we believe that this analysis (which would require the examination of a very large number of additional cysteine pairs) is beyond the scope of the present study. We did not examine simultaneous interactions between both seams because we were not convinced that such an analysis would provide any additional insights. After all, the data show that essentially all of the ^{MBP-76}EspP molecules contact BamA on *at least* one side through a β signal-BamA(β 1) interaction. Furthermore, we were concerned that the introduction of two cysteine substitutions into both ^{MBP-76}EspP and ^{His}BamA might create technical and/or conceptual problems that result from side reactions (e.g., the formation of intramolecular disulfide bonds, perhaps during passage of the proteins through the periplasm) or an inability to distinguish between singly and doubly disulfide-bonded adducts on SDS-PAGE.

3. Regarding the 'sliding' interpretation of the multiple interactions observed between EspP(β 1) along BamA(β 15/16): While I agree this is a reasonable interpretation, could an alternative be that these are not different steps in the process, but rather that there are just multiple solutions to the BamA(β 15) and EspP(β 1) interactions, any one of which could be productive and facilitate the swing? Also, if there is sliding, one could imagine that there is a potential for EspP(β 1) to interact with positions closer to the cell surface (e.g. BamA 789 or 801 or some other feature). The authors teased apart the interactions proximal to the periplasm, but, unless I missed it, there is still some space to explore potential interactions closer to the cell surface.

Strictly speaking, we cannot determine if we are observing distinct steps in the insertion process or multiple solutions to the interaction between BamA(β 15) and EspP(β 1). The difference between these two possibilities, however, may be very subtle. OMPs must ultimately end up in the lipid bilayer, and the degree to which they reach their destination by 'sliding' along the outward-facing surface of BamA(β 15/16) as opposed to entering spontaneously after making an initial contact with BamA(β 15/16) may in part be a matter of definition. Examining interactions between EspP(β 1) and BamA residues that are located closer to the cell surface would not clearly distinguish between the two scenarios or affect our conclusions, but would only indicate the length of the EspP(β 1)-BamA(β 15/16) interface.

4. A couple other minor points:

- Line 111. Might be convenient to refer to Figure 1b at the end of this sentence.

We have made the suggested change.

- Consider re-numbering the panels in Figure 2 (move the current 2E to 2C)

We have made the suggested change and also switched Figs. 2D and 2E so that they are presented in the same order as they are mentioned in the text.

- Line 232. Is there a possibility that an interface that is not apparent from the available structures (which are solved in the absence of substrate) could be present when substrate is engaged and that was missed? Are there any other positions to consider for interaction?

It is certainly conceivable that there is an additional interface that is not apparent from the available structures, but we might have to make an enormous number of BamA/EspP cysteine pairs in order to find it. Given that we would be stabbing in the dark with no guarantee of success, we believe that an effort to find additional interfaces is beyond the scope of the present study.

- Line 379. Although not the simplest explanation, one could also imagine that there are different intermediates for different OMPs (e.g., autotransporters versus OMPs with lipoproteins bound in lumen versus trimeric OMPs versus simple OMPs, etc.). This is the first and only example of an intermediate being trapped on BAM so it will be interesting to see, in the future, how different substrates interact with this complex.

The reviewer raises a good point here. Although we agree that it should be very interesting to determine how other OMPs interact with the BamA β -barrel, we discuss evidence that the swing model applies to many (or all) bacterial OMPs in the last paragraph of the Discussion. In light of the reviewer's comment, we have now modified lines 408-409 to indicate more clearly that

studies conducted on several different OMPs *in vivo* favor the swing model over the threading model.

Reviewer #2:

The manuscript by Doyle and Bernstein describes the structural mapping of a blocked intermediate of the insertase protein BamA inserting a suitably modified version of the substrate protein EspP. Spatial proximities in the blocked intermediate state are identified by chemical cross-linking via engineered disulfide bridges and analyzed by Western blots. They authors find a well-defined interface between strands b1 of BamA and b12 of the substrate and a less-well defined interface with apparently multiple conformations between b1 of EspP and b15/b16 of BamA. The work is very well done, the data are of outstanding quality and everything is well documented and described, along with sufficient control experiments. The demonstration of the existence of a stable intermediate of BamA-EspP and the derivation of some structural information for this protein complex is a significant contribution to understanding Bam and its mechanism. This is a nice piece of work and I clearly support publication in Nat. Comm.

Thanks for the positive feedback!

The following points should however be addressed before publication.

1.) The present work was done with the autotransporter EspP as a client and extrapolates from these data to all OMPs in general. The model shown in Figure 7 should therefore also include the situation of autotransporter clients and indicate at which point the passenger domains is expected to translocate to the extracellular side. I assume this must be at the hybrid barrel state (ii).

We agree with the reviewer and have modified the legend to Fig. 7 (lines 844-845) to address this concern. We found that we could capture ~40% of the ^{MBP-76}EspP molecules in a state in which both a loop of the passenger domain was exposed on the cell surface and the middle of strand $\beta 1$ of the β barrel (residue A1043) was positioned near the bottom of BamA strand $\beta 15$ (residue G781C) (Fig. 5b, 5c). Based on this observation, we think that at least the exposure of the passenger domain occurs prior to most of the swinging action. Our experiments do not, however, address the exact timing of the active translocation reaction. In the interest of proposing a general model for OMP assembly and keeping Fig. 7 as simple as possible, we have not included the passenger domain in the cartoon itself.

2.) The present work investigates the BamA-mediated insertion mechanism of an autotransporter. There has been a previous mechanistic model for autotransporter insertion by the insertase TamA, which is homologous to BamA (Gruss et al, NSMB 20, 1318 (2013)). The two resulting models are very similar: Steps ii-iv are essentially identical, they only differ in the question whether the barrel is pre-formed in the periplasm or not. The previous work needs to be credited and the authors should briefly compare the two mechanisms in the discussion.

We looked carefully at the paper by Gruss et al. and found that the authors propose a model for TamA function (based solely on structural data) that is very different from our model for BamA function. Their model is essentially a "threading" model in which the β barrel of the client protein is assembled within the plane of the outer membrane in a stepwise fashion between TamA β strands 1 and 16. Although in principle we would be happy to cite the paper,

we believe that it would be inaccurate and confusing to describe their model as similar to our swing model.

3.) In the model at stage (i), the client OMP is substantially curved, such that the hydrophobic side of the beta-sheet is concave. Such a shape does not really fit to the topology of the native membrane in three dimensions. It also requires that strand b1 is located somewhere in the membrane center, with a row of unsaturated hydrogen bond-donors and acceptors facing the hydrophobic phase of the bilayer. That is at least how it seems to be depicted in Figure 7. Such a conformation is energetically very unfavorable. The authors should explain in the text how exactly this is possible and/or improve the model stage (i) to be more realistic.

We agree that stage (i) is drawn in a somewhat confusing fashion, and we believe that the transparency of the client protein is a likely source of the confusion. A concave hydrophobic surface would indeed be unrealistic and is the opposite of the conformation that we intended to depict. After consulting with a graphic artist in our institution, we have tried to improve the entire Figure by adding a solid outline to BamA and the client protein, and by modifying some of the shading. It should now be clearer that the hydrophobic surface of the client protein is convex. In the interest of keeping the cartoon as consistent with the data as possible, we have also drawn $\beta 1$ of the client OMP closer to the outward-facing surface of BamA $\beta 15/16$.

4.) In the title and elsewhere, the authors refer to the transition complex as "asymmetric". With this, the authors want to express that the interfaces b1/b12 and b1/b15/b16 are structurally not equivalent. The chosen wording may cause confusion, since a heterodimeric protein complex is a priori always asymmetric. Perhaps a different wording can be found.

While we agree with the reviewer that the term "asymmetric" is imperfect, we have struggled to find a better descriptor that captures the striking difference between the two intra-barrel interfaces that is concise. A couple of alternatives we considered were "skewed" and "unequal", but these and other terms seem even more problematic. All things considered, we believe that "asymmetric" conveys our meaning most clearly and represents the best option. Nevertheless, to address the reviewer's concern, we have modified the text to define the term "asymmetric hybrid-barrel" when we introduce it for the first time on line 85. We now state that "We found that the BamA β -barrel formed two dissimilar interfaces with the assembly intermediate that effectively creates a *structure that we refer to as an 'asymmetric hybrid-barrel'.*"

Reviewer #3:

Summary. The biogenesis of beta-barrel outer membrane proteins (OMPs) is still poorly understood, despite the structures of the individual components of BAM being reported, along with recent X-ray and cryoEM structures of fully assembled BAM. It is thought that BAM modulates the membrane bilayer in order to allow the spontaneous or orchestrated insertion of OMPs. Exactly how the OMPs are able to make their way to BAM and the for biogenesis are still elusive. The authors here use EspP in a crosslinking study of a stalled intermediate to decipher how BAM interacts with EspP during folding into the OM.

Comments (in no particular order):

1. Overall, the paper is well written and organized and presents studies on the biogenesis of OMPs by BAM. The crosslinking methods are a strength of the paper and provide a wealth of information about the interaction of EspP with BAM.

Many thanks for the positive feedback!

2. The author refer to the interactions as ‘asymmetric’ interactions, even in the title, yet it is a bit unclear what this refers to as opposed to a ‘symmetric’ interaction? Could the author elaborate on this, as none of the barrels are symmetric not would one expect a ‘symmetric’ interaction given this. Possible remove such jargon for clarity?

See our reply to a similar concern raised by reviewer 2 (comment 4).

3. Line 26 – ‘open’ form of BamA, is this the barrel open to the surface or the POTRA open? Or just that the lateral gate is open? For an interaction to occur with the exposed edge of beta1 of BamA, it would be assumed that the it is a lateral gate in open state.

To address this concern, we now state that the BamA β barrel is “laterally open”.

4. Line 27 – why ‘Unexpectedly?’ This study is the first to interrogate such a system, and the results are all new. Maybe it was meant in contrast to the Sam50 study? And if so, it is still not surprising that there are differences between the two systems. Further, this study is based on much more convincing structures than the Sam50 study and therefore easier to interpret.

We have replaced “unexpectedly” with “in contrast”.

5. Line 85 – ‘asymmetric hybrid-barrel’, again, why is asymmetric needed here?

Once again, please see our reply to reviewer 2 (comment 4).

6. Lines 86-87 – maybe it was missed, but no data seemed to be presented either in favor or against a ‘threading’ model, as only the N-term and C-term strand were studied primarily and assuming these possibly interact first, one would have to interrogate within the client OMP before concluding anything about a ‘threading’ mechanism or not.

We present two lines of evidence that strongly disfavor the “threading” model. First, while the threading model predicts that client proteins would face the inner surface of the BamA β barrel, our data show that the first strand of the EspP β barrel faces the outer surface of the BamA β barrel and does not form disulfide bonds with a lumen-facing control (V784C) when the assembly of ^{MBP-76}EspP stalls (Fig. 4 and Supplementary Fig. 7). Second, we observed strong disulfide bond formation between EspP A1043C, a residue located near the middle of the fully folded β barrel, and BamA G781C, a residue located near the periplasm. The proximity of these two residues indicates that much of the first β strand of the EspP β barrel is situated in the periplasm when assembly stalls. This configuration is inconsistent with a threading model that predicts the transfer of client protein β strands from the lumen of the BamA β barrel directly into the plane of the outer membrane. Furthermore, based on our data we can speculate but cannot conclude that the C-terminal strand of the EspP β barrel is the first segment of the client protein that interacts with the BamA β barrel.

7. EspP is an autotransporter and therefore likely has unique properties and biogenesis from other OMPs. While this doesn’t deter from these nice studies, the authors should make this a clear point since the barrel is mostly assembled and just awaiting maturation upon cleavage. This is not like other OMPs that don’t have a passenger domain or cleavage activity. Therefore, it is likely that BAM interactions with these class a bit differently in order to allow the passenger biogenesis.

Based on the information that we have provided about autotransporters (see p. 6), we believe that it should be clear that we are analyzing a late stage in assembly. A priori, we do not see a clear reason to think that the presence of the passenger domain and the intra-barrel cleavage reaction (which may occur after the β barrel is fully assembled and released from BAM) per se suggest that autotransporter β barrels are assembled in a fundamentally different fashion from other OMP β barrels. In this regard it should be noted that a significant part of our study involves examining the interaction between BamA and the β signal—a highly conserved segment that is found in most bacterial OMPs. Furthermore, as mentioned in our reply to reviewer 1 (comment 4, part 4), we describe evidence from previous studies that suggests that a swing mechanism is used in the assembly of many other OMPs in the last paragraph of the Discussion.

8. Line 156 and Fig 1d/e – the levels in panel d appear very faint however are much more obvious in panel e, what was the different here?

We are not sure which bands the reviewer is referring to. There were differences in the protocols used to generate the data in panels d and e (see the legend to Fig. 1 and Methods), but the levels of most of the bands are comparable. In fact, the intensity of some of the key bands (PK fragments, mature EspP β barrel) are higher in panel d than panel e.

9. Line 176 – ‘fold completely’ suggests that BamA is not folded properly and possibly dysfunctional. However, the lack of the disulfide is more likely just making the loop more dynamic and therefore more accessible to PK, which is much different than having a mis-folded BamA. Authors should consider revising this for clarity, esp for a more general audience. With the disulfide, the loop is more compact, less dynamic, and less accessible for PK digestion.

We have modified the text to address this concern. We now state that the cysteine residues are needed for loop 6 to fold into a native conformation.

10. Throughout the manuscript, the authors refer to the ‘outer surface’ but this is a bit confusing since it could mean towards the surface side of the bacteria, or the outside of the barrel. Authors should consider making sure this is clear what is meant throughout each portion of the manuscript.

To improve clarity we have changed ‘outer surface’ to ‘outward-facing surface’ throughout.

11. Page 16, the authors make the claim that differing conformational states are observed for one crosslink while not for the other. While the gel does appear to show smearing or laddering, one should be cautious of pre-boil samples as this can also be a function of stability also. For example, one may form a tight interaction while the other forms a more loose interaction that slow breaks down as the gel is ran. If these are truly different conformations, PK digests might be able to tease this out and be a nice complement here.

We are a bit confused by this comment. Gel-based mobility-shift assays that are similar to those shown in Fig. 6c have been used for many years to study differences in β barrel protein conformations. The multiple forms of $^{MBP-76}EspP_{S1299C}-^{His}BamA_{S425C}$ that we observe cannot be due to degradation because they are detected by antibodies against the N and C termini of both BamA and the client protein. Indeed as the reviewer seems to suggest, the simplest interpretation of the results (which is essentially the interpretation we provide) is that after $^{MBP-76}EspP_{S1299C}$ and $^{His}BamA_{S425C}$ undergo disulfide bond formation, the opposing interface interacts more “loosely” than the opposing interface of the $^{MBP-76}EspP_{A1043C}-^{His}BamA_{G781C}$ pair. The PK

digests that the reviewer recommends would not facilitate detection of multiple conformational states because they would cleave the exposed ^{MBP-76}EspP passenger domain loop and thereby alter the structure of the BamA-client protein complex in unpredictable ways.

12. The crosslinking studies are nicely done and the authors make a very convincing case for the interaction of bamA beta-1 and espP beta-12, and the same for the beta-1 to some degree. However, the resulting proposed 'hinge' mechanism is much less convincing as noted below:

a. The authors are weighing the model based almost entirely on the observations of the beta1 interaction of EspP with BamA, but this is likely the final before insertion by strand exchange. The authors don't appear to consider the initiating step here of beta-12 which seems to be the most stable and wouldn't appear to have the nearly parallel to the membrane conformation as suggested in the model.

As stated in the Discussion (lines 390-392) and illustrated in Fig. 7 (step i) and Video 1, we do think that the interaction between EspP(β 12) and BamA(β 1) is the initiating step. Furthermore, to maintain consistency with the orientation of BamA(β 1) in the crystal structure of BAM, we initially depict the BamA(β 1)-EspP(β 12) interface at a $\sim 45^\circ$ angle to the plane of the membrane—not parallel to the membrane. At a later stage (step ii) we depict the BamA(β 1)-EspP(β 12) interface more perpendicular to the membrane to maintain consistency with solved structures that show a remarkable strand rotation in BamA.

b. The suggestion that one face of the client OMP faces the periplasm and then inverts itself and concavity seems highly unlikely. While the hydrophilic face would be facing the periplasm, the hydrophobic face would be completely exposed to the surface of the bacteria or the inner side of the membrane, both of which are not likely. Further, what is the force or mechanism by which this conversion would be regulated or mediated, as the protein(s) would not just adopt this transition automatically?

Please see our reply to reviewer 1 (comment 1, part 2) and reviewer 2 (comment 3).

c. What is the role of the other Bam components? This mechanism is BamA centric, what would the other Bam proteins be doing here to mediate OMP biogenesis?

Please see our reply to reviewer 1 (first part of comment 1).

d. The model depends on a rigid motion of BamA rotating the entire client OMP into the membrane, which would require quite a bit of torque and energy, where would this come from?

Please see our reply to reviewer 1 (second part of comment 1).

e. Overall, this reviewer is less convinced by the proposed model despite being excited about the rest of the manuscript. One could just as convincingly argue that autotransporters undergo a mechanism sharing features of both major models and the data would align equally as well as for the new proposal model. The current model isn't fully supported by the literature or by the studies presented and only raises more questions than piecing things together; therefore, suggest removing the model to strengthen the overall manuscript.

We do not agree that it would be possible to devise a convincing explanation of our data by combining the two leading models. The specific interactions between BamA and the ^{MBP-76}EspP assembly intermediate that we observed effectively rule out the idea that BAM catalyzes OMP assembly simply by perturbing the lipid bilayer ('assisted' model). Furthermore, as stated in our

response to comment 6, our results strongly challenge the 'threading' model. In order to explain our data, it is really necessary to construct a new model. Although we certainly agree with the reviewer that many questions remain unanswered, we believe that our model fits well not only with the results presented in our manuscript but also with the results of previous studies.

Reviewers' Comments:

Reviewer #1:

Remarks to the Author:

Doyle and Bernstein have provided reasonable responses to all queries and their additions/changes to the manuscript help to clarify questions that were raised. The described work is complete, will provide important data for multiple fields, and should spark the interest of the broad audience of Nature Communications. This reviewer continues to find this work valuable and enthusiastically supports its publication in Nature Communications.

Reviewer #2:

Remarks to the Author:

Doyle and Bernstein have addressed most issues appropriately, except point 2: Especially if the authors find that their model is partially different from the model previously suggested by Gruss et al, this needs to be discussed in the discussion section. Both models propose formation of a hybrid barrel as an insertion intermediate and towards a constructive discussion in the field, it is essential to assess, how (and perhaps why) the differences manifest.

Once this issue has been clarified, I strongly support publication of the work.

Reviewer #3:

None

Response to referees

REVIEWERS' COMMENTS:

Reviewer #1 (Remarks to the Author):

Doyle and Bernstein have provided reasonable responses to all queries and their additions/changes to the manuscript help to clarify questions that were raised. The described work is complete, will provide important data for multiple fields, and should spark the interest of the broad audience of Nature Communications. This reviewer continues to find this work valuable and enthusiastically supports its publication in Nature Communications.

We thank the reviewer for his/her strong endorsement!

Reviewer #2 (Remarks to the Author):

Doyle and Bernstein have addressed most issues appropriately, except point 2: Especially if the authors find that their model is partially different from the model previously suggested by Gruss et al, this needs to be discussed in the discussion section. Both models propose formation of a hybrid barrel as an insertion intermediate and towards a constructive discussion in the field, it is essential to assess, how (and perhaps why) the differences manifest.

Once this issue has been clarified, I strongly support publication of the work.

We have now cited the paper by Gruss et al. in the Discussion (end of second paragraph) and have indicated that their model posits a 'threading' mechanism that is distinct from the 'swing' mechanism that we propose.